# LEARNING INVARIANT REPRESENTATIONS OF TIME-HOMOGENEOUS STOCHASTIC DYNAMICAL SYSTEMS

**Vladimir R. Kostic**[*]
Istituto Italiano di Tecnologia
University of Novi Sad
vladimir.kostic@iit.it

**Pietro Novelli**[*]
Istituto Italiano di Tecnologia
pietro.novelli@iit.it

**Riccardo Grazzi**
Istituto Italiano di Tecnologia
University College London

**Karim Lounici**
CMAP École Polytechnique

**Massimiliano Pontil**
Istituto Italiano di Tecnologia
University College London

## ABSTRACT

We consider the general class of time-homogeneous stochastic dynamical systems, both discrete and continuous, and study the problem of learning a representation of the state that faithfully captures its dynamics. This is instrumental to learning the transfer operator or the generator of the system, which in turn can be used for numerous tasks, such as forecasting and interpreting the system dynamics. We show that the search for a good representation can be cast as an optimization problem over neural networks. Our approach is supported by recent results in statistical learning theory, highlighting the role of approximation error and metric distortion in the learning problem. The objective function we propose is associated with projection operators from the representation space to the data space, overcomes metric distortion, and can be empirically estimated from data. In the discrete-time setting, we further derive a relaxed objective function that is differentiable and numerically well-conditioned. We compare our method against state-of-the-art approaches on different datasets, showing better performance across the board.

## 1 INTRODUCTION

Dynamical systems are a mathematical framework describing the evolution of state variables over time. These models, often represented by nonlinear differential equations (ordinary or partial) and possibly stochastic, have broad applications in science and engineering, ranging from climate sciences (Cannarsa et al., 2020; Fisher et al., 2009), to finance (Pascucci, 2011), to atomistic simulations (Mardt et al., 2018; McCarty and Parrinello, 2017; Schütte et al., 2001), and to open quantum system dynamics (Gorini and Kossakowski, 1976; Lindblad, 1976), among others. However, it is usually the case that no analytical models of the dynamics are available and one must resort to data-driven techniques to characterize a dynamical system. Two powerful paradigms have emerged: deep neural networks (DNN) and kernel methods. The latter are backed up by solid statistical guarantees (Steinwart and Christmann, 2008) determining when linearly parameterized models can be learned efficiently. Yet, selecting an appropriate kernel function may be a hard task, requiring a significant amount of experimentation and expertise. In comparison, the former are very effective in learning complex data representations (Goodfellow et al., 2016), and benefit from a solid ecosystem of software tools, making the learning process feasible on large-scale systems. However, their statistical analysis is still in its infancy, with only a few proven results on their generalization properties.

Kernel-based methods hinge on the powerful idea of characterizing dynamical systems by lifting their definition over a Hilbert space of functions and then studying the associated *transfer operators*. They describe the average evolution of functions of the state (*observables*) over time and for deterministic systems are also known as *Koopman operators*. Furthermore, transfer operators are *linear*, and under additional assumptions admit a spectral decomposition, which provides a valuable tool to interpret and analyze the behavior of non-linear systems (see e.g. Brunton et al., 2022; Kutz et al., 2016, and

---

[*]Equal contribution, corresponding authors.

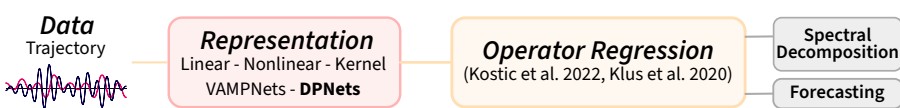

Figure 1: Pipeline for learning dynamical systems. DPNets learn a data representation to be used with standard operator regression methods. In turn, these are employed to solve downstream tasks such as forecasting and interpreting dynamical systems via spectral decomposition.

references therein). The usefulness of this approach critically relies on an appropriate choice of the *observable space*. In particular, in order to fully benefit from the spectral decomposition, it is crucial to find an observable space $\mathcal{F}$ which linearizes the dynamics and is *invariant* under the action of the transfer operator. In kernel-based methods, such a space is implicitly linked to the kernel function and gives a clear mathematical meaning to the problem of choosing a "good" kernel. Unfortunately, the analytical form of transfer operators is often intractable or unavailable, especially in complex or poorly understood systems, posing challenges in constructing invariant representations.

In this paper, we build synergistically upon both kernel and DNN paradigms: we first employ DNNs to learn an invariant representation that fully captures the system dynamics, and then forward this representation to kernel-based algorithms for the actual transfer operator regression task. This general framework is illustrated in Fig. 1. Our method, named Deep Projection Networks (DPNets), addressing the challenge of providing good representations to the operator regression algorithms, can be cast as an optimization problem over neural networks and can benefit from a differentiable and numerically well-conditioned score functional, enhancing the stability of the training process.

**Previous work** Extensive research has been conducted on learning dynamical systems from data. The monographs (Brunton et al., 2022; Kutz et al., 2016) are standard references in this field. To learn transfer operators we mention the works (Alexander and Giannakis, 2020; Bouvrie and Hamzi, 2017; Das and Giannakis, 2020; Kawahara, 2016; Klus et al., 2019; Kostic et al., 2022; Williams et al., 2015b) presenting kernel-based algorithms, and (Azencot et al., 2020; Bevanda et al., 2021; Fan et al., 2021; Lusch et al., 2018; Morton et al., 2018) based on deep learning schemes. Finding meaningful representations of the state of the system to be used in conjunction with transfer operator learning is a critical challenge, tackled by many authors. We mention the works (Azencot et al., 2020; Federici et al., 2024; Lusch et al., 2018; Morton et al., 2018; Otto and Rowley, 2019) where a representation is learned via DNN schemes, as well as (Kawahara, 2016; Li et al., 2017; Mardt et al., 2019; 2018; Tian and Wu, 2021; Yeung et al., 2019; Wu and Noé, 2019) addressing the problem of learning invariant subspaces of the transfer operator. Mostly related to our methodology is (Andrew et al., 2013), which introduced deep canonical correlation analysis and, especially, VAMPnets (Mardt et al., 2018; Wu and Noé, 2019), which repurposed this approach to learn transfer operators. Within our theoretical framework, indeed, we will show that VAMPNets can be recovered as a special case in the setting of discrete dynamics. More in general, operator learning with DNN approaches is reviewed e.g. in Kovachki et al. (2023), with a specific focus on PDEs.

**Contributions** Our contributions are: **1)** Leveraging recent results in statistical learning theory, we formalize the problem of representation learning for dynamical systems and design a score function based on the orthogonal projections of the transfer operator in data spaces (Sec. 2); **2)** We show how to reliably solve the corresponding optimization problem. In the discrete case, our score leads to optimal representation learning under the mild assumption of compactness of the transfer operator (Thm. 1). In the continuous case, our method applies to time reversal invariant dynamics (Thm. 2) including the important case of Langevin dynamics; **3)** We show how the learned representation can be used within the framework of operator regression in a variety of settings. Numerical experiments illustrate the versatility and competitive performance of our approach against several baselines.

**Notation** We let $\mathbb{N}$ be the set of natural numbers and $\mathbb{N}_0 = \{0\} \cup \mathbb{N}$. For $m \in \mathbb{N}$ we denote $[m] := \{1, \dots, m\}$. If $\mathcal{T}$ is a compact linear operator on two Hilbert spaces we let $\sigma_i(\mathcal{T})$ be its $i$-th largest singular value, we let $(\mathcal{T})^\dagger$ be the Moore-Penrose pseudo-inverse, and $\mathcal{T}^*$ the adjoint operator. We denote by $\|\cdot\|$ and $\|\cdot\|_{\mathrm{HS}}$ the operator and Hilbert-Schmidt norm, respectively.

## 2 REPRESENTATION LEARNING FOR DYNAMICAL SYSTEMS

In this section, we give some background on transfer operators for dynamical systems, discuss key challenges in learning them from data, and propose our representation learning approach.

**Background** Let $(X_t)_{t\in\mathbb{T}}$ be a stochastic process taking values in some state space $\mathcal{X}$, where the time index $t$ can be either discrete ($\mathbb{T}=\mathbb{N}_0$) or continuous ($\mathbb{T}=[0,+\infty)$), and denote by $\mu_t$ the law of $X_t$. Let $\mathcal{F}\subset\mathbb{R}^{\mathcal{X}}$ be a prescribed space of real-valued functions, henceforth referred to as *observables space*. By letting $s,t\in\mathbb{T}$, with $s\leq t$, the *forward transfer operator* $\mathcal{T}_{s,t}\colon\mathcal{F}\to\mathcal{F}$ evolves an observable $f:\mathcal{X}\to\mathbb{R}$ from time $s\in\mathbb{T}$ to time $t$, by the conditional expectation

$$[\mathcal{T}_{s,t}(f)](x) := \mathbb{E}[f(X_t)\,|\,X_s=x],\ x\in\mathcal{X}. \tag{1}$$

For a large class of stochastic dynamical systems, these *linear* operators are *time-homogeneous*, that is they only depend on the time difference $t-s$. In this case $\mathcal{T}_{s,t}=\mathcal{T}_{0,t-s}=:\mathcal{T}_{t-s}$. Further, we will use the shorthand notation $\mathcal{T}:=\mathcal{T}_1$. Time-homogeneous transfer operators at different times are related through the Chapman-Kolmogorov equation (Allen, 2007), $\mathcal{T}_{s+t}=\mathcal{T}_t\,\mathcal{T}_s$, implying that the family $(\mathcal{T}_t)_{t\in\mathbb{T}}$ forms a semigroup. The utility of transfer operators is that, when the space $\mathcal{F}$ is suitably chosen, they *linearize* the process. Two key requirements are that $\mathcal{F}$ is *invariant* under the action of $\mathcal{T}_t$, that is $\mathcal{T}_t[\mathcal{F}]\subseteq\mathcal{F}$ for all $t$ (a property that we tacitly assumed above), and *rich enough* to represent the flow of the process. Two common choices for $\mathcal{F}$ that fulfill both requirements are the space of bounded Borel-measurable functions and the space of square-integrable functions $L^2_\pi(\mathcal{X})$ with respect to the invariant distribution $\pi$, when one exists. To ease our presentation we first focus on the latter case, and then extend our results to non-stationary time-homogeneous processes. Formally, the process $(X_t)_{t\in\mathbb{T}}$ admits an *invariant distribution* $\pi$ when $\mu_s=\pi$ implies $\mu_t=\pi$ for all $s\leq t$ (Ross, 1995). This in turn allows one to define the transfer operator on $\mathcal{F}=L^2_\pi(\mathcal{X})$.

While in the discrete-time setting the whole process can be studied only through $\mathcal{T}=\mathcal{T}_1$, when time is continuous the process is characterized by the infinitesimal generator of the semigroup $(\mathcal{T}_t)_{t\geq 0}$, defined as

$$\mathcal{L} := \lim_{t\to 0^+}(\mathcal{T}_t - I)/t. \tag{2}$$

$\mathcal{L}$ can be properly defined on the Sobolev space $W^{1,2}_\pi(\mathcal{X})\subset L^2_\pi(\mathcal{X})$ formed by functions with square-integrable gradients (see Lasota and Mackey, 1994; Ross, 1995).

**Learning transfer operators** In practice, dynamical systems are only observed, and neither $\mathcal{T}$ nor its domain $\mathcal{F}=L^2_\pi(\mathcal{X})$ are known, providing a key challenge to learn them from data. The most popular algorithms (Brunton et al., 2022; Kutz et al., 2016) aim to learn the action of $\mathcal{T}\colon\mathcal{F}\to\mathcal{F}$ on a predefined Reproducing Kernel Hilbert Space (RKHS) $\mathcal{H}$, forming a subset of functions in $\mathcal{F}$. This allows one, via the kernel trick, to formulate the problem of learning the *restriction* of $\mathcal{T}$ to $\mathcal{H}$, $\mathcal{T}_{|\mathcal{H}}\colon\mathcal{H}\to\mathcal{F}$, via empirical risk minimization (Kostic et al., 2022). However, recent theoretical advances (Korda and Mezić, 2017; Klus et al., 2019; Nüske et al., 2023), proved that such algorithms are statistically consistent only to $P_\mathcal{H}\mathcal{T}_{|\mathcal{H}}$, where $P_\mathcal{H}$ is the orthogonal projection onto the closure of $\mathcal{H}$ in $\mathcal{F}$. The projection $P_\mathcal{H}$ constrains the *evolved* observables back inside $\mathcal{H}$, thereby, in general, altering the dynamics of the system. Therefore, to assure that one properly learns the dynamics, two major requirements on $\mathcal{H}$ are needed: i) $\mathcal{T}_{|\mathcal{H}}$ needs to approximate well $\mathcal{T}$, i.e. the space $\mathcal{H}$ needs to be big enough relative to the domain of $\mathcal{T}$; ii) the difference between the projected restriction and the true one, i.e. the approximation error $\big\|[I-P_\mathcal{H}]\mathcal{T}_{|\mathcal{H}}\big\|$, needs to be small.

When $\mathcal{H}$ is an infinite-dimensional *universal* RKHS, both the above requirements are satisfied (Kostic et al., 2022), i.e. $\mathcal{H}$ is dense in $\mathcal{F}$ and the approximation error is zero, leading to an arbitrarily good approximation of dynamics with enough data. Still, another key issue arises: the norms on the a-priori chosen $\mathcal{H}$ and the unknown $\mathcal{F}$ do not coincide, since the latter depends on the process itself. This *metric distortion* phenomenon has been recently identified as the source of spurious estimation of the spectra of $\mathcal{T}$ (Kostic et al., 2023), limiting the utility of the learned transfer operators. Indeed, even if $\mathcal{T}$ is self-adjoint, that is the eigenfunctions are orthogonal in $\mathcal{F}$, the estimated ones will not be orthogonal in $\mathcal{H}$, giving rise to spectral pollution (Kato, 1976). This motivates one to additionally require that iii) $\mathcal{H}$ is a *subspace* of $\mathcal{F}$, i.e. both spaces have the same norm.

To summarize, the desired optimal $\mathcal{H}$ is the *leading invariant subspace* of $\mathcal{T}$, that is the subspace corresponding to the largest (in magnitude) eigenvalues of $\mathcal{T}$. This subspace $\mathcal{H}$ achieves zero approximation error, eliminates metric distortion and best approximates (in the dynamical system sense) the operator $\mathcal{T}$. Since any RKHS $\mathcal{H}$ is entirely described by a *feature map*, learning a leading invariant subspace $\mathcal{H}$ from data is, fundamentally, a *representation learning* problem.

**Approach** We start by formalizing the problem of learning a good finite dimensional representation space for $\mathcal{T}$, and then address the same for the generator $\mathcal{L}$. We keep the discussion less formal here, and state our main results more precisely in the next section. Our approach is inspired by the

following upper and lower bounds on the approximation error, a direct consequence of the norm change from $\mathcal{H}$ to $\mathcal{F}$,

$$\left\|[I - P_{\mathcal{H}}]\mathcal{T}P_{\mathcal{H}}\right\|^2 \lambda_{\min}^+(C_{\mathcal{H}}) \leq \left\|[I - P_{\mathcal{H}}]\mathcal{T}_{|\mathcal{H}}\right\|^2 \leq \left\|[I - P_{\mathcal{H}}]\mathcal{T}P_{\mathcal{H}}\right\|^2 \lambda_{\max}(C_{\mathcal{H}}), \qquad (3)$$

where $C_{\mathcal{H}}$ is the covariance operator on $\mathcal{H}$ w.r.t. the measure $\pi$, while $\lambda_{\min}^+$ and $\lambda_{\max}$ are the smallest and largest non-null eigenvalues, respectively. Note that the norms on the hypothetical domain $\mathcal{H}$ and true domain $L_\pi^2(\mathcal{X})$ coincide if and only if $C_{\mathcal{H}} = I$, in which case equalities hold in (3) and the approximation error becomes $\left\|[I - P_{\mathcal{H}}]\mathcal{T}P_{\mathcal{H}}\right\|$.

When the operator $\mathcal{T}$ is known, the latter quantity can be directly minimized by standard numerical algorithms for spectral computation to find invariant subspaces (see e.g. Golub and Van Loan, 2013). Unfortunately, in our stochastic setting $\mathcal{T}$ is unknown since we cannot compute the conditional expectation in (1). To overcome this issue we propose a learning approach to recover the invariant space $\mathcal{H}$, which is rooted in the singular value decomposition, holding under the mild assumption that $\mathcal{T}$ is a compact operator[1]. The main idea is that the subspace made of the leading $r$ *left singular functions* of $\mathcal{T}$ serves as a good approximation of the desired leading invariant subspace of $\mathcal{T}$. Namely, due to the orthonormality of the singular functions, we have that $C_{\mathcal{H}} = I$ and $P_{\mathcal{H}}\mathcal{T}$ becomes the $r$-truncated SVD of $\mathcal{T}$, that is, its best rank-$r$ approximation. Therefore, according to (3), the approximation error is at most $\sigma_{r+1}(\mathcal{T})$, which can be made arbitrarily small by rising $r$. Moreover, the distance of the subspace of left singular functions to the desired leading invariant subspace is determined by the "*normality*" of $\mathcal{T}$ (Trefethen and Embree, 2020). If the operator $\mathcal{T}$ is *normal*, that is $\mathcal{T}\mathcal{T}^* = \mathcal{T}^*\mathcal{T}$, then both its left and right singular spaces are invariant subspaces of $\mathcal{T}$, resulting in zero approximation error irrespectively of $r$. This leads us to the following optimization problem

$$\max_{\mathcal{H}, \mathcal{H}' \subset L_\pi^2(\mathcal{X})} \left\{ \left\|P_{\mathcal{H}}\mathcal{T}P_{\mathcal{H}'}\right\|_{\mathrm{HS}}^2 \mid C_{\mathcal{H}} = C_{\mathcal{H}'} = I, \dim(\mathcal{H}) \leq r, \dim(\mathcal{H}') \leq r \right\}. \qquad (4)$$

Relying on the application of Eckart-Young-Mirsky's Theorem (Kato, 1976), we can show that the desired representation space $\mathcal{H}$ can be computed by solving (4). Note that, in general, the auxiliary space $\mathcal{H}'$ is needed to capture right singular functions, while if we have prior knowledge that $\mathcal{T}$ is normal without loss of generality one can set $\mathcal{H} = \mathcal{H}'$ in (4).

The same reasoning cannot be applied straight away to the generator $\mathcal{L}$ of a continuous dynamical system, which in is not even guaranteed to be bounded (Lasota and Mackey, 1994), let alone compact. For *time-reversal invariant* processes, however, $\mathcal{T}$ and $\mathcal{L}$ are self-adjoint, that is $\mathcal{T} = \mathcal{T}^*$ and $\mathcal{L} = \mathcal{L}^*$. This includes the important case of Langevin dynamics, which are of paramount importance for molecular dynamics (see, e.g., Tuckerman, 2010). For such systems we show (see Theorem 2) that the leading $r$-dimensional subspace of the generator $\mathcal{L}$ can be found by solving the following optimization problem featuring the *partial trace* objective

$$\max_{\mathcal{H} \subset W_\pi^{1,2}(\mathcal{X})} \left\{ \mathrm{tr}\left(P_{\mathcal{H}}\,\mathcal{L}\,P_{\mathcal{H}}\right) \mid C_{\mathcal{H}} = I, \dim(\mathcal{H}) \leq r \right\}, \qquad (5)$$

where now the projector is $P_{\mathcal{H}} \colon W_\pi^{1,2}(\mathcal{X}) \to W_\pi^{1,2}(\mathcal{X})$. We stress that, since we assume $\mathcal{L} = \mathcal{L}^*$, we can relax the above assumption on the compactness of the transfer operators $(\mathcal{T}_t)_{t \geq 0}$, and still show that the leading invariant subspace of $\mathcal{L}$ is the optimal $\mathcal{H}$ in (5). So, on such $\mathcal{H}$ we can estimate well $\mathcal{L}$ via generator operator regression, see e.g. (Hou et al., 2023; Klus et al., 2020).

## 3 DEEP PROJECTION SCORE FUNCTIONALS

In this section we show how to solve the representation learning problems (4) and (5) using DNNs. In the discrete case, we consider a generalized version of problem (4), which encompasses non-stationary processes, for which the probability distributions change along the trajectory. Namely, let $X$ and $X'$ be two $\mathcal{X}$-valued random variables with probability measures $\mu$ and $\mu'$, respectively. Because of time-homogeneity, w.l.o.g. $X$ models the state at time 0 and $X'$ its evolution after some time $t$. Then, the transfer operator can be defined on the data-dependent domains $\mathcal{T}_t \colon L_{\mu'}^2(\mathcal{X}) \to L_\mu^2(\mathcal{X})$. Replacing $P_{\mathcal{H}'}$ with $P_{\mathcal{H}'}' \colon L_{\mu'}^2(\mathcal{X}) \to L_{\mu'}^2(\mathcal{X})$ in (4), and using the covariances $C_{\mathcal{H}}$ and $C_{\mathcal{H}'}$ w.r.t.

---

[1]This property is fulfilled by a large class of Markov processes (see e.g. Kostic et al., 2022) and is weaker than requiring the operator being Hilbert-Schmidt as in (Mardt et al., 2018).

the measures $\mu$ and $\mu'$, we obtain the appropriate representation learning problem for non-stationary process; see App. B for more details. Within this general setting we optimize two feature maps

$$\psi_w : \mathcal{X} \to \mathbb{R}^r, \text{ and } \psi'_w : \mathcal{X} \to \mathbb{R}^r, \tag{6}$$

parameterized by $w$ taking values in some set $\mathcal{W}$. Next, defining the two RKHSs[2] $\mathcal{H}_w :=$ $\mathrm{span}(\psi_{w,j})_{j \in [r]}$ and $\mathcal{H}'_w := \mathrm{span}(\psi'_{w,j})_{j \in [r]}$, both equipped with the standard inner product, we aim to solve (4) by optimizing over the weights $w$. To avoid solving the constrained optimization problem, we further propose to relax the hard constraints in (4) through a *metric distortion loss* $\mathcal{R} : \mathbb{R}^{r \times r} \to \mathbb{R}_+$ that is zero if and only if it is applied to the identity matrix. Our choice in Section 5, well defined for SPD matrices $C$, is

$$\mathcal{R}(C) := \mathrm{tr}(C^2 - C - \ln(C)).$$

Finally, in Lem. 3, App. C we show that (4) can be solved by maximizing, for $\gamma > 0$

$$\mathcal{P}^\gamma(w) = \left\| (C_X^w)^{\dagger/2} C_{XX'}^w (C_{X'}^w)^{\dagger/2} \right\|_{\mathrm{HS}}^2 - \gamma \left( \mathcal{R}(C_X^w) + \mathcal{R}(C_{X'}^w) \right), \tag{7}$$

where we introduced the uncentered covariance matrices

$$C_X^w := \mathbb{E}\, \psi_w(X)\psi_w(X)^\top, \; C_{X'}^w := \mathbb{E}\, \psi'_w(X')\psi'_w(X')^\top \text{ and } C_{XX'}^w := \mathbb{E}\, \psi_w(X)\psi'_w(X')^\top. \tag{8}$$

Notice that if $\gamma = 0$ and the covariances $C_X^w$ and $C_{X'}^w$ are nonsingular, then the score reduces to $\left\| (C_X^w)^{-\frac{1}{2}} C_{XX'}^w (C_{X'}^w)^{-\frac{1}{2}} \right\|_{\mathrm{HS}}^2$ which is called VAMP-2 score in the VAMPNets approach of (Mardt et al., 2018). Moreover, if in (7) the HS norm is replaced by the nuclear norm, the score becomes the objective of canonical correlation analysis (CCA) (Harold, 1936; Hardoon et al., 2004) in feature space. When DNNs are used to parametrize (6), such score is the objective of Deep-CCA (Andrew et al., 2013), known as VAMP-1 score (Mardt et al., 2018) in the context of molecular kinetics. Therefore, by maximizing $\mathcal{P}^\gamma$ we look for the strongest *linear correlation* between $\mathcal{H}_w$ and $\mathcal{H}'_w$. We stress that differently from Deep-CCA and VAMPNets the crucial addition of $\mathcal{R}$ is to *decorrelate* the features within each space, guiding the maximization towards a solution in which the norms of $\mathcal{H}$ and $\mathcal{H}'$ coincide with those of $L_\mu^2(\mathcal{X})$ and $L_{\mu'}^2(\mathcal{X})$, overcoming metric distortion.

While in the optimal representation the covariances $C_X^w$ and $C_{X'}^w$ are the identity, in general they are non-invertible, making the score $\mathcal{P}^\gamma$ non-differentiable during optimization. Indeed, unless the rank of both covariances is stable for every $w \in \mathcal{W}$, we might have exploding gradients (Golub and Pereyra, 1973). Even ignoring differentiability issues, the use of the pseudo-inverse, as well as the use of the inverse in the non-singular case, can introduce severe numerical instabilities when estimating $\mathcal{P}^\gamma$ and its gradients during the training process. More precisely, the numerical conditioning of evaluating $\mathcal{P}^\gamma$ using (7) is determined by the covariance condition numbers $\lambda_1(C_X^w)/\lambda_r(C_X^w)$ and $\lambda_1(C_{X'}^w)/\lambda_r(C_{X'}^w)$ that can be very large in practical situations (see e.g. the fluid dynamics example in Sec. 5) To overcome this issue, we introduce the relaxed score

$$\mathcal{S}^\gamma(w) := \left\| C_{XX'}^w \right\|_{\mathrm{HS}}^2 / \left( \left\| C_X^w \right\| \left\| C_{X'}^w \right\| \right) - \gamma \left( \mathcal{R}(C_X^w) + \mathcal{R}(C_{X'}^w) \right), \tag{9}$$

which, as implied by Theorem 1 below, is a lower bound for the score in (7). A key advantage of this approach is that the score $\mathcal{S}^\gamma$ is both differentiable (apart from the trivial case $C_X^w, C_{X'}^w = 0$) and has stable gradients, since we avoid matrix inversion. Indeed, computing $\mathcal{S}^\gamma$ is always well-conditioned, as the numerical operator norm has conditioning equal to one.

The following theorem provides the theoretical backbone of our representation learning approach.

**Theorem 1.** *If $\mathcal{T} : L_{\mu'}^2(\mathcal{X}) \to L_\mu^2(\mathcal{X})$ is compact, $\mathcal{H}_w \subseteq L_\mu^2(\mathcal{X})$ and $\mathcal{H}'_w \subseteq L_{\mu'}^2(\mathcal{X})$, then for all $\gamma \geq 0$*

$$\mathcal{S}^\gamma(w) \leq \mathcal{P}^\gamma(w) \leq \sigma_1^2(\mathcal{T}) + \cdots + \sigma_r^2(\mathcal{T}). \tag{10}$$

*Moreover, if $(\psi_{w,j})_{j \in [r]}$ and $(\psi'_{w,j})_{j \in [r]}$ are the leading $r$ left and right singular functions of $\mathcal{T}$, respectively, then both equalities in (10) hold. Finally, if the operator $\mathcal{T}$ is Hilbert-Schmidt, $\sigma_r(\mathcal{T}) > \sigma_{r+1}(\mathcal{T})$ and $\gamma > 0$, then the "only if" relation is satisfied up to unitary equivalence.*

This result establishes that the relaxed score is a tight lower bound, in the optimization sense, for (7). More precisely, in general $|\mathcal{S}^\gamma(w) - \mathcal{P}^\gamma(w)|$ is arbitrarily small for small enough $\mathcal{R}(w)$, while on the feasible set in (4) both scores coincide. For instance, for a deterministic linear system this happens as

---

[2]Here, $\psi_{w,j}$ and $\psi'_{w,j}$ are the $j$-th component of $\psi_w$ and $\psi'_w$, respectively.

---

**Algorithm 1** DPNets Training

---

**Input:** Data $\mathcal{D}$; metric loss parameter $\gamma$; DNNs $\psi_w, \psi'_w : \mathcal{X} \to \mathbb{R}^r$; optimizer $U$; # of steps $k$.
1: Initialize DNN weights to $w_1$.
2: **for** $j = 1$ to $k$ **do**;
3:      $\hat{C}_X^{w_j}, \hat{C}_{X'}^{w_j}, \hat{C}_{XX'}^{w_j} \leftarrow$ Covariances for $\psi_{w_j}$ and $\psi'_{w_j}$ from a mini-batch of $m \le n$ samples.
4:      $F(w_j) \qquad\qquad \leftarrow -\widehat{\mathcal{S}}_m^\gamma(w_j)$ from (9) and the covariances in Step 3.
5:      $w_{j+1} \qquad\qquad \leftarrow U(w_j, \nabla F(w_j))$ where $\nabla F(w_j)$ is computed using backpropagation.
6: **end for**
7: **return** representations $\psi_{w_K}, \psi'_{w_K}$.

---

soon as the classes of features $\psi_w$ and $\psi'_w$ include linear functions. In general, we can appeal to the universal approximation properties of DNN (Cybenko, 1989) to fulfill the hypotheses of Theorem 1.

**Generator learning** We now consider how to learn the representation of continuous time-homogeneous processes arising from stochastic differential equations (SDE) of the form

$$dX_t = A(X_t)\,dt + B(X_t)\,dW_t, \tag{11}$$

where $A$ and $B$ are the drift and diffusion terms, respectively, and $W_t$ is a Wiener process. The solution of (11) is a stochastic process that is time-reversal invariant w.r.t. its invariant distribution $\pi$, (Arnold, 1974). To show that solving (5) leads to learning the leading invariant subspace of $\mathcal{L}$, we restrict to the case $\mathcal{L} = \mathcal{L}^*$ and consider the partial trace of $\mathcal{L}$ w.r.t. to the subspace $\mathcal{H}_w := \mathrm{span}(\psi_{w,j})_{j\in[r]} \subseteq W_\pi^{1,2}(\mathcal{X})$ as our objective function. Being $\mathcal{L}$ self-adjoint, we can relax the compactness assumption of Thm. 1, requiring only the much weaker condition that $\mathcal{L}$ has the largest eigenvalues separated from the *essential spectrum* (Kato, 1976). In particular, it holds $\mathrm{tr}\,(P_{\mathcal{H}_w}\,\mathcal{L}\,P_{\mathcal{H}_w}) = \mathrm{tr}\,((C_X^w)^\dagger C_{X\partial}^w)$, where $C_{X\partial}^w = \mathbb{E}[\psi_w(X)\,d\psi_w(X)^\top]$ is the continuous version of the cross-covariance and $d\psi_w(\cdot)$ is given by the Itō formula (see e.g. (Arnold, 1974; Klus et al., 2020; Hou et al., 2023))

$$d\psi_{w,i}(x) := \mathbb{E}[\nabla\psi_{w,i}(X)^\top dX/dt + \tfrac{1}{2}(dX/dt)^\top \nabla^2\psi_{w,i}(X)(dX/dt)\,|\,X = x] \tag{12}$$

$$= \nabla\psi_{w,i}(x)^\top A(X) + \tfrac{1}{2}\,\mathrm{tr}(B(x)^\top \nabla^2\psi_{w,i}(x)B(x)). \tag{13}$$

The following result, the proof of which is given in App. E, then justifies our continuous DPNets.

**Theorem 2.** *If $\mathcal{H}_w \subseteq W_\pi^{1,2}(\mathcal{X})$, and $\lambda_1(\mathcal{L}) \ge \cdots \ge \lambda_{r+1}(\mathcal{L})$ are eigenvalues of $\mathcal{L}$ above its essential spectrum, then for every $\gamma \ge 0$ it holds*

$$\mathcal{P}_\partial^\gamma(w) := \mathrm{tr}\,\big((C_X^w)^\dagger C_{X\partial}^w\big) - \gamma\mathcal{R}(C_X^w) \le \lambda_1(\mathcal{L}) + \cdots + \lambda_r(\mathcal{L}), \tag{14}$$

*and the equality is achieved when $\psi_{w,j}$ is the eigenfunction of $\mathcal{L}$ corresponding to $\lambda_j(\mathcal{L})$, for $j \in [r]$.*

Notice how $\mathrm{tr}\,\big((C_X^w)^\dagger C_{X\partial}^w\big)$ in Eq. (14) is the sum of the (finite) eigenvalues of the symmetric eigenvalue problem $C_{X\partial}^w - \lambda C_X^w$. Thus, in non-pathological cases, its value and gradient can be computed efficiently in a numerically stable way, see e.g. (Andrew and Tan, 1998).

## 4 METHODS

**Learning the representation** To optimize a DPNet representation one needs to replace the population covariances and cross-covariance in Eqs (7), (9) and (14) with their empirical counterparts. In practice, given $\psi_w, \psi'_w : \mathcal{X} \to \mathbb{R}^r$ and a dataset $\mathcal{D} = (x_i, x'_i)_{i\in[n]}$ consisting of samples from the joint distribution of $(X, X')$ one estimates the needed covariances $C_X^w, C_{X'}^w$ and $C_{XX'}^w$ as

$$\widehat{C}_X^w = \tfrac{1}{n}\sum_{i\in[n]}\psi_w(x_i)\psi_w(x_i)^\top,\ \widehat{C}_{X'}^w = \tfrac{1}{n}\sum_{i\in[n]}\psi'_w(x'_i)\psi'_w(x'_i)^\top \text{ and } \widehat{C}_{XX'}^w = \tfrac{1}{n}\sum_{i\in[n]}\psi_w(x_i)\psi'_w(x'_i)^\top.$$

In App. D.1 we prove that the empirical scores $\widehat{\mathcal{P}}_n^\gamma : \mathcal{W} \to \mathbb{R}$ and $\widehat{\mathcal{S}}_n^\gamma : \mathcal{W} \to \mathbb{R}$, that is the scores computed using empirical covariances, concentrate around the true ones with high probability for any fixed $w$. Obtaining uniform guarantees over $w$ requires advanced tools from empirical processes and regularity assumptions on the representations which could be the subject of future work. The difficulties arising from the estimation of $\mathcal{P}^\gamma$ during optimization are also addressed.

In Alg. 1 we report the training procedure for DPNets-relaxed (using score (9)), which can also be applied to (7) and (14) suitably modifying step 4. When using mini-batch SGD methods to compute $\nabla \widehat{\mathcal{P}}_m^\gamma(w)$ or $\nabla \widehat{\mathcal{S}}_m^\gamma(w)$, the batch size $m$ should be taken sufficiently large to mitigate the impact of biased stochastic estimators. The time complexity of the training algorithm is addressed in App. D.4.

**Operator regression** Once a representation $\psi_w$ is learned, it can be used within the framework of operator regression, see Fig. 1. Following the reasoning in (Kostic et al., 2022) one sees that any model $\widehat{\mathcal{T}}_w \colon \mathcal{H}_w \to \mathcal{H}_w$ of the transfer operator $\mathcal{T}$ acts on functions in $\mathcal{H}_w$ as

$$h_z := \psi_w(\cdot)^\top z \quad \mapsto \quad \widehat{\mathcal{T}}_w h_z := \psi_w(\cdot)^\top \widehat{T} z, \; z \in \mathbb{R}^r, \tag{15}$$

where $\widehat{T} \in \mathbb{R}^{r \times r}$ is a matrix in the representation space. For example, denoting the data matrices $\widehat{\Psi}_w = [\psi_w(x_1) \,|\, \cdots \,|\, \psi_w(x_n)]$, $\widehat{\Psi}'_w = [\psi_w(x'_1) \,|\, \cdots \,|\, \psi_w(x'_n)] \in \mathbb{R}^{r \times n}$, the ordinary least square estimator (OLS) minimizes the empirical risk $\left\| \widehat{\Psi}'_w - \widehat{T}^\top \widehat{\Psi}_w \right\|_{\mathrm{HS}}^2$, yielding $\widehat{T} := (\widehat{\Psi}_w^\top)^\dagger \widehat{\Psi}_w'^\top$.

**Downstream tasks** Once $\widehat{T}$ is obtained, it can be used predict the next expected state of the dynamics given the initial one, compute the modal decomposition of an observable (Kutz et al., 2016), estimate the spectrum of the transfer operator (Kostic et al., 2023) or controlling the dynamics (Proctor et al., 2016). Indeed, recalling that $X'$ is the one step ahead evolution of $X$, we can use (15) to approximate $\mathbb{E}[h_z(X') \,|\, X = x]$ as $\psi_w(x)^\top \widehat{T} z$. Moreover, relying on the reproducing property of $\mathcal{H}_w$, the predictions can be extended to functions $f \colon \mathcal{X} \to \mathbb{R}^\ell$ as $\mathbb{E}[f(X') \,|\, X = x] \approx \psi_w(x)^\top (\widehat{\Psi}_w^\top)^\dagger \widehat{F}'$, where $\widehat{F}' = [f(x'_1) \,|\, \ldots \,|\, f(x'_n)]^\top \in \mathbb{R}^{n \times \ell}$ is the matrix of the observations of the evolved data. Clearly, when the observable is the state itself (i.e. $f(x) = x$), we obtain one step ahead predictions.

Next, observe that eigenvalue decomposition of the matrix $\widehat{T}$ leads to the eigenvalue decomposition of the operator $\widehat{\mathcal{T}}$. Namely, let $(\widehat{\lambda}_i, \widehat{u}_i, \widehat{v}_i) \in \mathbb{C} \times \mathbb{C}^d \times \mathbb{C}^d$ be an eigen-triplet made of eigenvalue, left eigenvector and right eigenvector of $\widehat{T}$, that is $\widehat{T}\widehat{v}_i = \widehat{\lambda}_i \widehat{v}_i$, $\widehat{u}_i^* \widehat{T} = \widehat{\lambda}_i \widehat{u}_i^*$ and $\widehat{u}_i^* \widehat{v}_k = \delta_{i,k}$, $i, k \in [r]$, we directly obtain, using (15), the spectral decomposition

$$\widehat{\mathcal{T}}_w = \sum_{i \in [r]} \widehat{\lambda}_i \, \widehat{f}_i \otimes \widehat{g}_i, \quad \text{where} \quad \widehat{f}_i(x) := \psi_w(x)^\top \widehat{v}_i \;\text{ and }\; \widehat{g}_i(x) := (\widehat{u}_i)^* \psi_w(x). \tag{16}$$

Finally, we can use the spectral decomposition of $\widehat{T}$ to efficiently forecast observables for several time-steps in the future via $\mathbb{E}[h_z(X_t) \,|\, X_0 = x] \approx \sum_{i \in [r]} \widehat{\lambda}_i^t \, \langle \widehat{g}_i, h_z \rangle_{\mathcal{H}_w} \, \widehat{f}_i(x)$, which is known as the extended dynamic mode decomposition (EDMD), see e.g. (Brunton et al., 2022). Noticing that $\langle \widehat{g}_i, h_z \rangle_{\mathcal{H}_w} = (\widehat{u}_i)^* z = (\widehat{u}_i)^* \widehat{T} z / \widehat{\lambda}_i$, when $\widehat{\lambda}_i \neq 0$, and, again, using the reproducing property, we can extend this approximation to vector-valued observables $f \colon \mathcal{X} \to \mathbb{R}^\ell$ as

$$\mathbb{E}[f(X_t) \,|\, X_0 = x] \approx \sum_{i \in [r]} \widehat{\lambda}_i^{t-1} \left( \psi_w(x)^\top \widehat{v}_i \right) \left( \widehat{u}_i^* (\widehat{\Psi}_w^\top)^\dagger \widehat{F}' \right) \in \mathbb{R}^\ell. \tag{17}$$

## 5 EXPERIMENTS

In this experimental section we show that DPNets (i) learn reliable representations of non-normal dynamical systems, (ii) can be used in its continuous-dynamics form (14) to approximate the eigenvalues of a Langevin generator, and (iii) outperforms kernel methods and auto-encoders in large-scale and/or structured (e.g. graphs and images) data settings. To have a fair comparison, every neural network model in these experiments has been trained on the same data splits, batch sizes, number of epochs, architectures and seeds. The learning rate, however, has been optimized for each one separately. We defer every technical detail, as well as additional results to App. F. The code to reproduce the examples can be found at `https://pietronvll.github.io/DPNets/`, and it heavily depends on Kooplearn `https://kooplearn.readthedocs.io/`.

**Baselines** We compared our methods DPNets ($\mathcal{P}^\gamma$) and DPNets-relaxed ($\mathcal{S}^\gamma$), where appropriate, with the following baselines: Dynamic AE (DAE) of (Lusch et al., 2018), Consistent AE of (Azencot et al., 2020), DMD of (Schmid, 2010), KernelDMD and ExtendedDMD of (Williams et al., 2015a;b), and VAMPNets of (Mardt et al., 2018).

**Logistic Map** We study the dynamical system $X_{t+1} = (4X_t(1 - X_t) + \xi_t) \bmod 1$, for $\mathcal{X} = [0, 1)$ and $\xi_t$ being i.i.d. trigonometric noise (Ostruszka et al., 2000) . The associated transfer operator $\mathcal{T}$ is *non-normal*, making the learning of its spectral decomposition particularly challenging (Kostic et al., 2023). Since $\mathcal{T}$ can be computed exactly, we can sidestep the problem of operator regression (see

| Representation | Spectral Error | Optimality Gap |
| --- | --- | --- |
| **DPNets** | 0.28 (0.03) | **0.64 (0.01)** |
| **DPNets-relaxed** | **0.06 (0.05)** | 1.19 (0.04) |
| VAMPNets | 0.21 (0.09) | 0.97 (0.22) |
| Cheby-T | 0.20 | 1.24 |
| NoiseKernel | 0.19 | 2.17 |

Table 1: *Logistic Map*. Comparison of DPNets to relevant baselines. Mean and standard deviation are over 20 independent runs. We used $r = 7$.

| Model | $\mathcal{P}$ | Transition | Enthalpy $\Delta H$ |
| --- | --- | --- | --- |
| **DPNets** | **12.84** | **17.59 ns** | **-1.97 kcal/mol** |
| Nys-PCR | 7.02 | 5.27 ns | -1.76 kcal/mol |
| Nys-RRR | 2.22 | 0.89 ns | -1.44 kcal/mol |
| | | | |
| Reference | - | 40 ns | -6.1 kcal/mol |

Table 2: *Chignolin*. Comparison between DP-Nets and kernel methods with Nyström sampling (Meanti et al., 2023).

Fig. 1), and focus directly on evaluating the quality of the representation encoded by $\mathcal{H}_w$. We evaluate DPNets on two different metrics: (i) the optimality gap $\sum_{i=1}^{r} \sigma_i^2(\mathcal{T}) - \mathcal{P}^0(w)$ for $r = 3$ and (ii) the spectral error, given by $\max_i \min_j |\lambda_i(P_{\mathcal{H}} \mathcal{T}_{|\mathcal{H}}) - \lambda_j(\mathcal{T})|$. While (i) informs on how close one is to solve (4), (ii) measures how well the true eigenvalues of $\mathcal{T}$ can be recovered within the representation space $\mathcal{H}_w$. In Tab. 1 we compare the DPNets representation against VAMPNets, ExtendedDMD with a feature map of Chebyshev polynomials and the feature defining the trigonometric noise Ostruszka et al. (2000) of the process itself. Notice that in this challenging setting recovering eigenvalues via singular vectors is generally problematic, so a moderate optimality gap may lead to a larger spectral error. In Fig. 2 we report the evolution of the spectral error during training for DPNets and VAMPNets, while we defer to App. F.1 an in-depth analysis of the role of the feature dimension. Notice that DPNets and DPNets-relaxed excel in both metrics.

**Continuous dynamics** We investigate a one-dimensional *continuous* SDE describing the stochastic motion of a particle into the Schwantes potential (Schwantes and Pande, 2015). The invariant distribution for this process is the Boltzmann distribution $\pi(dx) \propto e^{-\beta V(x)} dx$, where $V(x)$ is the potential at state $x$. The non-null eigenvalues of $\mathcal{L}$ hold physical significance, as their absolute value represents the average rate at which particles cross one of the system's potential barriers (Kramers, 1940); our objective is to accurately estimate them. Here $X_t$ is sampled *non-uniformly* according to a geometric law. In the lower panel of Fig. 2 we report the estimated transition rates, of $\mathcal{L}$ along the DPNets training loop. Notice that the embedding $\psi_w$ progressively improves eigenvalue estimation, indicating that the invariant subspaces of the generator $\mathcal{L}$ are well captured.

**Ordered MNIST** Following Kostic et al. (2022), we create a stochastic dynamical system by randomly sampling images from the MNIST dataset according to the rule that $X_t$ should be an image of the digit $t \pmod 5$ for all $t \in \mathbb{N}_0$. Given an image from the dataset with label $c$, a model for the transfer operator $\mathcal{T}$ of this system should then be able to produce an MNIST-alike image of the next digit in the cycle. In the upper panel of Fig. 3 we thus evaluate DPNets and a number of baselines by how accurate is an "oracle" supervised MNIST classifier (test accuracy for in-distribution $\geq 99\%$) in predicting the correct label $c + t \pmod 5$ after $t$ steps of evolution. DPNets consistently retain an accuracy above $95\%$, while for every other method it degrades. The "reference" line corresponds to random guessing, while the "Oracle-Feature" baseline is an operator regression model (EDMD) using, as the dictionary of functions, the output logits of the oracle, and despite having been trained with the true labels, its performance degrades drastically after $t \geq 5$.

**Fluid dynamics** We study the classical problem of the transport of a passive scalar field by a 2D fluid flow past a cylinder (Raissi et al., 2020). Each data point comprises a regular 2D grid that encompasses fluid variables, including velocity, pressure, and scalar field concentration at each grid point. This system is non-stationary and is also known to exhibit non-normal dynamics (see Trefethen and Embree, 2020). We evaluate each trained baseline by feeding to it the last snapshot of the train trajectory and evaluating the relative RMSE (that is, the RMSE normalized by the data variance) between the forecasts and the subsequent test snapshots. In this experiment, we also have a *physics-informed* (PINN) baseline not related to transfer operator learning, which is however the model for which this dataset was created in the first place. Remarkably, the forecasting error of DPNets does not grow sensibly with time as it does for every other method.

**The Metastable states of Chignolin** In our last experiment, we study the dynamics of Chignolin, a folding protein, from a $106\,\mu$s long molecular dynamics simulation sampled every $200\,$ps, totalling over $500{,}000$ data points (Lindorff-Larsen et al., 2011). We focus on the leading eigenfunctions of the transfer operator, which are known (Schütte et al., 2001) to provide a simple characterization of the slowest (and usually most important) physical processes occurring along the dynamics. From the leading left eigenfunctions of $\widehat{\mathcal{T}}$, one can indeed construct the free energy surface (see Fig. 4),

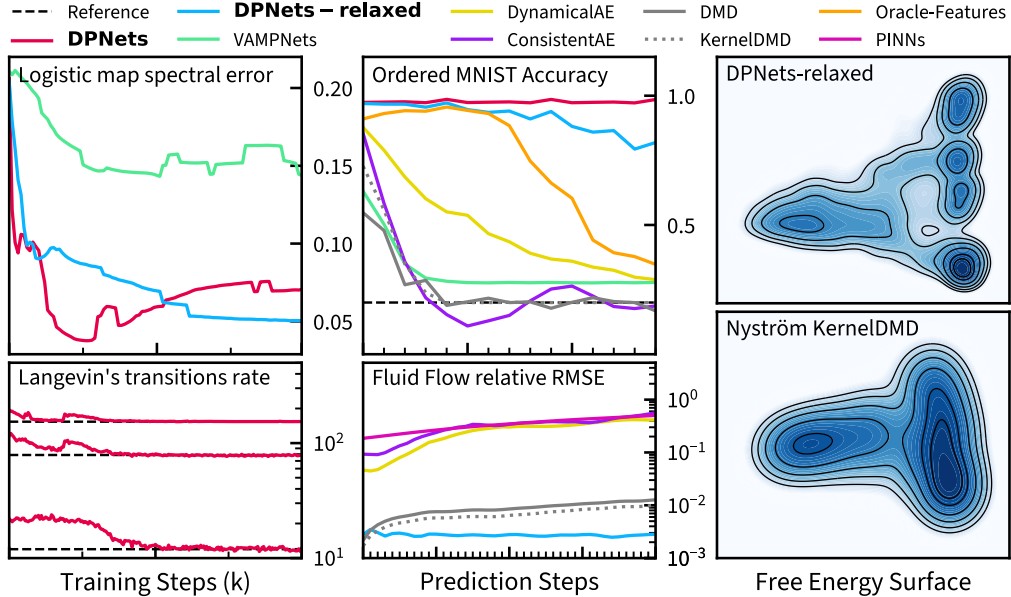

Figure 2: *Eigenvalue error decays during training*. Upper: spectral error for the logistic map. Lower: DPNets-estimated eigenvalues of $\mathcal{L}$ for the Langevin dynamics.

Figure 3: *Forecasting with DPNets*. Upper: classification accuracy over time for ordered MNIST. Lower: forecasting RMSE over time for fluid dynamic example.

Figure 4: Free energy surface of the 2 slowest modes of Chignolin, estimated by DP-Nets and Nyström PCR. To be compared with Bonati et al. (2021).

whose local minima identify the metastable states of the system. The free energy surface, indeed, is proportional to the negative $\log$-pdf of finding the system in a given state while at thermodynamic equilibrium, meaning that a state of *low* free energy is highly probable, hence metastable. For Chignolin, the metastable states have been thoroughly studied (see e.g. Novelli et al., 2022). The leading left eigenfunction of $\widehat{\mathcal{T}}$ encodes the folding-unfolding transition and the typical transition time is the *implied timescale* (Mardt et al., 2019) associated with its eigenvalue. The difference between the local minima of the free energy surface encodes the enthalpy $\Delta H$ of the transition. In Tab. 2 we compare these quantities to the reference values reported in Lindorff-Larsen et al. (2011). We trained a GNN-based DPNet-relaxed, as both DPNets unrelaxed and VAMPNets failed to converge, possibly due to the large scale of the data. We compared it to a KernelDMD estimator trained with the recent Nyström sketching technique (Meanti et al., 2023) as classical kernel methods are completely intractable at this scale[3]. Notice that DPNets succeed in finding additional meta-stable states of the system, which match the analysis of (Bonati et al., 2021); see App. F for more discussion.

# 6 CONCLUSIONS

We propose a framework for learning a representation of dynamical systems, based on orthogonal projections in data-spaces. It captures a leading invariant subspace of the transfer operator and can be applied to both discrete and continuous dynamical systems. In the discrete case, the representation is learned through the optimization of a smooth and numerically well conditioned objective function. Extensive numerical experiments demonstrate the effectiveness and generality of DPNets in various settings, suggesting that they are a promising tool for data-driven dynamical systems. A limitation of this work is that the score functional for the continuous systems might be unstable since it leverages covariance matrix inversion. Moreover, a future direction would be to study the statistical learning properties of the algorithm presented here.

---

[3] A back-of-the-envelope calculation shows that 450 GBs would be needed just to store kernel matrices in single precision.

## ACKNOWLEDGEMENTS

We acknowledge the financial support from the PNRR MUR Project PE000013 CUP J53C22003010006 "Future Artificial Intelligence Research (FAIR)", funded by the European Union – NextGenerationEU, EU Project ELIAS under grant agreement No. 101120237 and the chair "Business Analytic for Future Banking" sponsored by Natixis-Groupe BPCE.

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
