# Appendix

The appendix is organized as follows.

## A  BACKGROUND AND NOTATION

### A.1  STOCHASTIC PROCESSES TOOLS

Let $\mathcal{X}$ be a state space and $\Sigma_{\mathcal{X}}$ be the Borel $\sigma$-algebra on $\mathcal{X}$. Let $\mathcal{M}$ denote the space of all finite measures on $(\mathcal{X}, \Sigma_{\mathcal{X}})$. In what follows, we consider a random process $(X_t)_{t \in \mathbb{T}}$ with values in a state space $\mathcal{X}$.

**Definition 1** (Markov process.). *Let $\mathcal{F}_t := \Sigma\left(\{X_s : 0 \le s \le t\}\right)$ be the sigma-algebra generated by the elements of the process up to time $t$. The random process $\mathbf{X} = (X_t)_{t \in \mathbb{T}}$ is called a Markov process if for all $t, \tau \ge 0$ and $B \in \Sigma_{\mathcal{X}}$ it holds*

$$\mathbb{P}\left(X_{t+\tau} \in B | \mathcal{F}_t\right) = \mathbb{P}\left(X_{t+\tau} \in B | X_t\right), \tag{18}$$

*that is, conditioning on the full history of the process $\mathcal{F}_t$ up to time $t$ is equivalent to condition only on the state of the process at time $t$. For this reason, Markov process are sometimes called "memoryless".*

**Definition 2.** *The transition density function $p_\tau : \mathcal{X} \times \mathcal{X} \to [0, \infty]$ of a time-homogeneous process $\mathbf{X}$ is defined by*

$$\mathbb{P}\left(X_{t+\tau} \in B | X_t = x\right) = \int_B p_\tau(x, y) dy,$$

*for every measurable set $B \in \Sigma_{\mathcal{X}}$.*

The distribution of a time-homogeneous stochastic process $(X_t)_t$ with transition density functions $(p_\tau)_{\tau > 0}$ can be described by the semigroup of transfer operators $(\mathcal{T}_\tau)_{\tau \ge 0}$ usually defined on $L^\infty(\mathcal{X})$.

**Definition 3** (Transfer operator). *For any $\tau \ge 0$, the Koopman transfer operator $\mathcal{T}_\tau : L^\infty(\mathcal{X}) \to L^\infty(\mathcal{X})$ is defined by*

$$\mathcal{T}_\tau(f) = \mathbb{E}\left[f(X_{t+\tau}) | X_t = \cdot\right], \quad \forall f \in L^\infty(\mathcal{X}).$$

**Definition 4** (Feller process). *Let $(\mathcal{T}_\tau)_{\tau \ge 0}$ be the semigroup of transfer operators of a homogeneous Markov process. Then, $(\mathcal{T}_\tau)_{\tau \ge 0}$ is said to be a Feller semigroup when the following condition holds:*

1. *$\mathcal{T}_\tau\left(\mathcal{C}_0(\mathcal{X})\right) \subseteq \mathcal{C}_0(\mathcal{X})$ for all $\tau \ge 0$.*

2. *$\lim_{\tau \to 0} \|\mathcal{T}_\tau f - f\| = 0$ for all $f \in \mathcal{C}_0(\mathcal{X})$.*

*Here $\mathcal{C}_0(\mathcal{X})$ is the Banach space of all continuous functions vanishing at infinity.*

**Definition 5** (Infinitesimal generator of a semigroup). *Let $(\mathcal{T}_\tau)_{\tau \ge 0}$ be a Feller semigroup. We define its infinitesimal generator $\mathcal{L} : \mathcal{C}_0(\mathcal{X}) \to \mathcal{C}_0(\mathcal{X})$*

$$\mathcal{L}f = \lim_{\tau \to 0} \frac{1}{\tau}(\mathcal{T}_\tau f - f),$$

*for any $f \in \mathcal{C}_0(\mathcal{X})$ such that the above limit is well-defined.*

The above definitions can be lifted to $L^2_\pi(\mathcal{X})$ if the dynamical system has an invariant distribution, or if we are interested in the action of $\mathcal{T}_\tau$ on a specific couple of states $X, X'$ of the process, as discussed in the main text.

## A.2 Reproducing kernel Hilbert spaces

In this section we review few basic concepts on kernel based approaches to learning transfer operators; for more details on reproducing kernel Hilbert spaces (RKHS) we refer reader to (Aronszajn, 1950).

Let $k_X \colon \mathcal{X} \times \mathcal{X} \to \mathbb{R}$ and $k_{X'} \colon \mathcal{X} \times \mathcal{X} \to \mathbb{R}$ be two bounded kernels, and let $\mathcal{H}$ and $\mathcal{H}'$ be their respective reproducing kernel Hilbert spaces (RKHS). Denote the canonical feature maps by $\psi(x) := k_X(x, \cdot)$, $x \in \mathcal{X}$, and $\psi'(x') := k_{X'}(x', \cdot)$, $x' \in \mathcal{X}$.

Next, consider two probability measures $\mu$ and $\mu'$ on $\mathcal{X}$, and their associated $L^2$ spaces $L^2_\mu(\mathcal{X})$ and $L^2_{\mu'}(\mathcal{X})$. If $k_X(\cdot, \cdot)$ is square-integrable w.r.t. measure $\mu$ and $k_{X'}(\cdot, \cdot)$ is square-integrable w.r.t. the measure $\mu'$, then we can define injection (or the evaluation) operators $S_\mu \colon \mathcal{H} \hookrightarrow L^2_\mu(\mathcal{X})$ and $S_{\mu'} \colon \mathcal{H}' \hookrightarrow L^2_{\mu'}(\mathcal{X})$. Then, their adjoints $S_\mu^* \colon L^2_\mu(\mathcal{X}) \to \mathcal{H}$ and $S_{\mu'}^* \colon L^2_{\mu'}(\mathcal{X}) \to \mathcal{H}'$ are given by

$$S_\mu^* f = \int_\mathcal{X} f(x)\psi(x)\mu(dx) \in \mathcal{H} \quad \text{and} \quad S_{\mu'}^* f' = \int_\mathcal{X} f'(x')\psi'(x')\mu'(dx') \in \mathcal{H}',$$

where $f \in L^2_\mu(\mathcal{X})$ and $f' \in L^2_{\mu'}(\mathcal{X})$.

Using injections and their adjoints one can introduce covariance operators $C_X \colon \mathcal{H} \to \mathcal{H}$ and $C_{X'} \colon \mathcal{H}' \to \mathcal{H}'$ by

$$C_X := S_\mu^* S_\mu = \mathbb{E}_{X \sim \mu}\psi(X) \otimes \psi(X), \quad \text{and} \quad C_{X'} := S_{\mu'}^* S_{\mu'} = \mathbb{E}_{X' \sim \mu'}\psi(X') \otimes \psi(X'),$$

as well as the cross-covariance $C_{XX'} \colon \mathcal{H}' \to \mathcal{H}$ operator

$$C_{XX'} := S_\mu^* \mathcal{T} S_{\mu'} = \mathbb{E}_{(X,X') \sim \rho}\psi(X) \otimes \psi'(X'),$$

where $\rho$ is the joint measure of $(X, X')$ and $\mathcal{T} \colon L^2_{\mu'}(\mathcal{X}) \to L^2_\mu(\mathcal{X})$ is a transfer operator corresponding to the evolution of $X \sim \mu$ to $X' \sim \mu'$.

Different estimators for the problem of learning transfer operators from data $\mathcal{D} = (x_i, x_i')_{i \in [n]}$ have been considered, see e.g. (Kostic et al., 2022; Li et al., 2022), which are all of the form $\widehat{T}_W = \widehat{S}'^* W \widehat{S}$, where $W \in \mathbb{R}^{n \times n}$ and $\widehat{S} \colon \mathcal{H} \to \mathbb{R}^n$ and $\widehat{S}' \colon \mathcal{H}' \to \mathbb{R}^n$ are sampling operators given by $\widehat{S}f = n^{-\frac{1}{2}}[f(x_1) \ldots f(x_n)]^\top$, $f \in \mathcal{H}$, and $\widehat{S}'f' = n^{-\frac{1}{2}}[f'(x_1') \ldots f'(x_n')]^\top$, $f' \in \mathcal{H}'$. In particular, kernel methods usually consider *universal kernels* that generate infinite-dimensional RKHS spaces that are dense in $L^2$. In such a way, one can approximate transfer operators via operator (vector-valued) regression arbitrarily well with enough data. However, one important aspect has been recently reported in (Kostic et al., 2022; 2023). Namely, from the application perspective, utility of transfer operators is largely relying on the ability to estimate well their eigenvalues and eigenfunctions, which lead to the notion of modal decomposition of observables (Brunton et al., 2022). In that context, the difference in the norms between RKHS and $L^2$ spaces can produce spectral pollution and lead to bad estimation of the eigenvalues and eigenfunctions. We recall that the difference in norms is due to the covariance, i.e. for every $h \in \mathcal{H}$ its L2 norm is

$$\int_\mathcal{X} |h(x)|^2 \mu(dx) = \langle S_\mu h, S_\mu h \rangle_{L^2_\mu(\mathcal{X})} = \langle h, C_X h \rangle_\mathcal{H}. \tag{19}$$

While this difference in norms is unavoidable for universal kernels (indeed for universal kernels generating infinite-dimensional RKHS spaces, since covariance is a trace class operator, we have $\lambda_j(C_X) \to 0$ as $j \to \infty$ potentially leading to $\langle h, C_X h \rangle_\mathcal{H} \ll \langle h, h \rangle_\mathcal{H}$ for some $h \in \mathcal{H}$), for finite dimensional kernels one can hope to avoid metric distortion between spaces by aiming to find features for which $C_X = I$.

## B Representation Learning: problem and approach

In this section we prove some of the statements made in Sec. 2. In particular we show that (3) holds true, and show how the objectives (4) and (5) are related to the desired space $\mathcal{H}$.

We first note that $\mathcal{T}_{|\mathcal{H}} := \mathcal{T} S_\pi \colon \mathcal{H} \to L^2_\mu(\mathcal{X})$, while $\mathcal{T} P_\mathcal{H} = \mathcal{T} S_\pi C_X^\dagger S_\pi^*$. Hence, $\left\| [I - P_\mathcal{H}]\mathcal{T} P_\mathcal{H} \right\|^2 = \left\| [I - P_\mathcal{H}]\mathcal{T}_{|\mathcal{H}} C_X^{\dagger/2} \right\|^2 = \left\| C_X^{\dagger/2}([I - P_\mathcal{H}]\mathcal{T}_{|\mathcal{H}})^* \right\|^2$. Therefore, using that $\mathrm{Im}\left(([I - P_\mathcal{H}]\mathcal{T}_{|\mathcal{H}})^*\right) = \mathrm{Im}(S_\pi^* \mathcal{T}^*[I - P_\mathcal{H}]) \subseteq \mathrm{Im}(C_X^\dagger)$ (3) follows.

Next, we show how Eckart-Young's theorem justifies (4), which is the basis of Thm. 1.

**Lemma 1.** *Let $\mathcal{H}_1$ and $\mathcal{H}_2$ be two separable Hilbert spaces, and $A\colon \mathcal{H}_1 \to \mathcal{H}_2$ be a compact operator. Let $r \in \mathbb{N}$ and $\mathcal{P}_k^r := \{P\colon \mathcal{H}_k \to \mathcal{H}_k \mid P^* = P^2 = P, \operatorname{rank}(P) \le r\}$ denote the set of rank-$r$ orthogonal projectors in $\mathcal{H}_k$, $k = 1, 2$. It holds that*

*(i)* $\displaystyle \max_{(P_1,P_2)\in\mathcal{P}_1^r\times\mathcal{P}_2^r} \left\|P_2 A P_1\right\|_{\mathrm{HS}}^2 = \sum_{i\in[r]} \sigma_i^2(A),$

*(ii)* *Let $U_r \Sigma_r V_r^*$ be the $r$-truncated SVD of $A$, then $(V_r V_r^*, U_r U_r^*) \in$* $\displaystyle \arg\max_{(P_1,P_2)\in\mathcal{P}_1^r\times\mathcal{P}_2^r} \left\|P_2 A P_1\right\|_{\mathrm{HS}}^2.$

*(iii)* *If $\sigma_{r+1}(A) < \sigma_r(A)$ and $\left\|A\right\|_{\mathrm{HS}} < \infty$, then $\displaystyle \arg\max_{(P_1,P_2)\in\mathcal{P}_1^r\times\mathcal{P}_2^r} \left\|P_2 A P_1\right\|_{\mathrm{HS}}^2$ is singleton.*

*Proof.* This lemma is a direct consequence of the Eckart-Young theorem for compact operators (Kato, 1976). Recall, since $A$ is compact, there exists its SVD $A = U\Sigma V^*$ and Eckart-Young theorem states that if $A$ is Hilbert-Schmidt, for every $r \in \mathbb{N}$ and every $B\colon \mathcal{H}_1 \to \mathcal{H}_2$ such that $\operatorname{rank}(B) \le r$, it holds $\left\|A - B\right\|_{\mathrm{HS}} \ge \left\|A - A_r\right\|_{\mathrm{HS}}$, where $A_r := U_r \Sigma_r V_r^*$ denotes the $r$-truncated SVD of A. Moreover, if $\sigma_{r+1}(A) < \sigma_r(A)$, then equality implies $B = A_r$.

Hence, for arbitrary $P_k \in \mathcal{P}_k^r$, $k = 1, 2$, using that the norm of a projection is 1 and Pythagoras theorem, for every $m \ge r$ we have that

$$\left\|P_2 A_m P_1\right\|_{\mathrm{HS}}^2 \le \left\|P_2 A_m\right\|_{\mathrm{HS}}^2 \left\|P_1\right\|^2 = \left\|P_2 A_m\right\|_{\mathrm{HS}}^2 = \left\|A_m\right\|_{\mathrm{HS}}^2 - \left\|[I - P_2] A_m\right\|_{\mathrm{HS}}^2$$
$$= \left\|A_m\right\|_{\mathrm{HS}}^2 - \left\|A_m - P_2 A_m\right\|_{\mathrm{HS}}^2 \le \left\|A_m\right\|_{\mathrm{HS}}^2 - \left\|A_m - A_r\right\|_{\mathrm{HS}}^2 = \left\|A_r\right\|_{\mathrm{HS}}^2,$$

and, similarly, $\left\|P_2 A_m P_1\right\|_{\mathrm{HS}}^2 \le \left\|A_m\right\|_{\mathrm{HS}}^2 - \left\|A_m - A_m P_1\right\|_{\mathrm{HS}}^2 \le \left\|A_r\right\|_{\mathrm{HS}}^2$. However, since

$$\left| \left\|P_2 A P_1\right\|_{\mathrm{HS}} - \left\|P_2 A_m P_1\right\|_{\mathrm{HS}} \right| \le \left\|P_2 (A - A_m) P_1\right\|_{\mathrm{HS}} \le \sqrt{r}\left\|P_2 (A - A_m) P_1\right\|$$
$$\le \sqrt{r}\left\|A - A_m\right\| \le \sqrt{r}\sigma_{m+1}(A),$$

we obtain $\left\|P_2 A P_1\right\|_{\mathrm{HS}}^2 \le \left\|A_r\right\|_{\mathrm{HS}}^2 + \sqrt{r}\sigma_{m+1}(A)$ for all $m \ge r$. Thus, since $\left\|P_2 A P_1\right\|_{\mathrm{HS}}^2 = \left\|A_r\right\|_{\mathrm{HS}}^2$ obviously holds for $P_1 = V_r V_r^*$ and $P_2 = U_r U_r^*$, letting $m \to \infty$ we obtain (i) and (ii).

Finally, assume that $A$ is an Hilbert-Schmidt operator such that $0 \le \sigma_{r+1}(A) < \sigma_r(A)$. Then, similarly to the above now working with $A$ instead of $A_m$,

$$\left\|P_2 A P_1\right\|_{\mathrm{HS}}^2 \le \left\|A\right\|_{\mathrm{HS}}^2 - \max\{\left\|A - A P_1\right\|_{\mathrm{HS}}^2, \left\|A - P_2 A\right\|_{\mathrm{HS}}^2\} \le \left\|A_r\right\|_{\mathrm{HS}}^2.$$

So, if

$$(P_1, P_2) \in \arg\max_{(P_1',P_2')\in\mathcal{P}_1^r\times\mathcal{P}_2^r} \left\|P_2' A P_1'\right\|_{\mathrm{HS}}^2,$$

then

$$\max\{\left\|A - A P_1\right\|_{\mathrm{HS}}^2, \left\|A - P_2 A\right\|_{\mathrm{HS}}^2\} = \left\|A - A_r\right\|_{\mathrm{HS}}^2.$$

Now, the uniqueness result of the Eckart-Young theorem implies that $A P_1 = P_2 A = A_r$, and, consequently $\operatorname{rank}(P_1) = \operatorname{rank}(P_2) = r$, $A_r^\dagger A P_1 = A_r^\dagger A_r$ and $P_2 A A^\dagger = A_r A_r^\dagger$. Therefore, $V_r V_r^* P_1 = V_r V_r^*$ and $P_2 U_r U_r^* = U_r U_r^*$, imply $\operatorname{Im}(V_r) \subseteq \operatorname{Im}(P_1)$ and $\operatorname{Im}(U_r) \subseteq \operatorname{Im}(P_2)$. But since $P_1$ and $P_2$ have exactly rank $r$, we obtain $P_1 = V_r V_r^*$ and $P_2 = U_r U_r^*$. $\qquad\square$

We also discuss how to extend part of the analysis to the important setting of stationary deterministic systems for which the transfer operator is not compact.

**Remark 1.** *The compactness assumption on the the transfer operator, does not hold for purely deterministic dynamical systems. However, our approach is still applicable to the study of deterministic systems, that is when $X_{t+1} = F(X_t)$, for a deterministic map $F\colon \mathcal{X} \to \mathcal{X}$, and $X_t \sim \pi$ for all $t \in \mathbb{T}$, where $\pi$ is the invariant measure defined on the attractor. Then, we have that $\mathbb{E}[f(X_{t+1}) \mid X_t] = F \circ f$ and $\mathcal{T}\colon L_\pi^2(\mathcal{X}) \to L_\pi^2(\mathcal{X})$ is unitary, see e.g. (Brunton et al., 2022). Thus, $\mathcal{T}$ is not a compact operator, however it is a normal one. But then, Pythagoras theorem gives that*

$$\inf_{w\in\mathcal{W}} \left\|[I - P_{\mathcal{H}_w}] \mathcal{T} P_{\mathcal{H}_w}\right\|_{\mathrm{HS}}^2 = \inf_{w\in\mathcal{W}} \left( \left\|\mathcal{T} P_{\mathcal{H}_w}\right\|_{\mathrm{HS}}^2 - \left\|P_{\mathcal{H}_w} \mathcal{T} P_{\mathcal{H}_w}\right\|_{\mathrm{HS}}^2 \right) = r - \sup_{w\in\mathcal{W}} \mathcal{P}(w),$$

*where the second equality holds since the HS-norm is unitarily invariant. Therefore, when $w$ is such that $\mathcal{T}\psi_{w,j} = \lambda_j \psi_{w,j}$, since $\mathcal{T}$ is unitary we have $\mathcal{P}(w) = \sum_{j=1}^{r} |\lambda_j|^2 = r$ and the above inf is zero. Consequently, we have identified an $r$-dimensional invariant subspace of $\mathcal{T}$.*

Finally, we conclude this section with the lemma that is the basis of generator learning via optimization problem (5) whose objective is defined through a partial trace.

**Lemma 2.** *Let $\mathcal{H}$ be a separable Hilbert space and let $\mathcal{H}_r \subseteq \mathcal{H}$ be a (finite) $r$-dimensional subspace. If $A\colon \mathcal{H} \to \mathcal{H}$ is a self-adjoint operator having at least $r+1$ eigenvalues $\lambda_1(A) \geq \lambda_2(A) \geq \ldots \geq \lambda_r(A) \geq \lambda_{r+1}(A)$ above its essential spectrum, then*

$$\mathrm{tr}(AP_{\mathcal{H}_r}) \leq \lambda_1(A) + \lambda_2(A) + \cdots + \lambda_r(A),$$

*and equality holds when $\mathcal{H}_r$ is spanned by eigenfunctions of $A$ corresponding to eigenvalues $\lambda_1(A), \ldots, \lambda_r(A)$.*

*Proof.* First, let $(v_i)_{i \in [r]}$ denote eigenvectors of $A$ corresponding to eigenvalues $\lambda_i$, that is $Av_i = \lambda_i(A)v_i$, and let $(h_i)_{i \in \mathbb{N}}$ be an ortho-normal basis of $\mathcal{H}$ such that $(h_i)_{i \in [r]}$ is the ortho-normal basis of $\mathcal{H}_r$. Then, clearly

$$\mathrm{tr}(AP_{\mathcal{H}_r}) = \mathrm{tr}(P_{\mathcal{H}_r}AP_{\mathcal{H}_r}) = \sum_{i \in \mathbb{N}} \langle h_i, (P_{\mathcal{H}_r}AP_{\mathcal{H}_r})h_i \rangle = \sum_{i \in [r]} \langle h_i, Ah_i \rangle, \qquad (20)$$

which is clearly equal to $\lambda_1(A) + \lambda_2(A) + \ldots + \lambda_r(A)$ whenever $h_i = v_i$, $i \in [r]$.

The upper bound we prove by induction. First, for $r = 1$ we have that $\langle h_i, Ah_i \rangle \leq \lambda_1(A)$ follows directly from the Courant-Fischer max-min theorem for operators which claims that

$$\lambda_i(A) = \max_{h_1, \ldots, h_i} \min\{\langle h, Ah \rangle \mid h \in \mathrm{span}(h_j)_{j \in [i]}, \|h\| = 1\}, \ i \in [r+1]$$

Next, assuming that (20) holds for arbitrary $A$ and $r \leq m - 1$, we will prove that it holds for $r = m$.

Start by observing that there exists $\mathcal{H}' \subseteq \mathcal{H}_m$ such that $\dim(\mathcal{H}') \leq m - 1$ and $\mathcal{H}' \perp v_1$. Indeed, taking $g_i = \sum_{j \in [m]} b_{ij} h_j$, $i \in [m-1]$, for some $B = [b_{ij}] \in \mathbb{R}^{(m-1) \times m}$, we have that $g_i \perp v_1$ for all $i \in [m-1]$ iff $B\beta = 0$, where the vector $\beta \in \mathbb{R}^m$ is given by $\beta_j = \langle h_j, v_1 \rangle$. Now, if $\beta = 0$, then $\mathcal{H}_m \perp v_1$ and the claim trivially follows. Otherwise, since $\dim(\mathrm{span}(\beta)^\perp) = m - 1$, there exists a matrix $B \in \mathbb{R}^{(m-1) \times m}$ so that $B\beta = 0$. Consequently, the space $\mathcal{H}'$ spanned by $(g_i)_{i \in [m-1]}$ is such that $\dim(\mathcal{H}') \leq m - 1$ and $\mathcal{H}' \subseteq \mathrm{span}(v_1)^\perp$. Without the loss of generality, assume that $(g_i)_{i \in [m-1]}$ are orthonormal basis of $\mathcal{H}'$ and that $g_m$ is such that $(g_i)_{i \in [m]}$ are orthonormal basis of $\mathcal{H}$.

First, clearly $\langle g_m, Ag_m \rangle \leq \lambda_1$. On the other hand, define $A' := A + (\lambda_{r+1} - \lambda_1) v_1 \otimes v_1$. This is an operator obtained by deflating (moving) the first eigenvalue into the $(r+1)$-th one. Namely, we have that $\lambda_i(A') = \lambda_{i+1}(A)$ for $i \in [m-1]$. Moreover, it holds that $A'P_{\mathcal{H}'} = AP_{\mathcal{H}'}$. Hence, according to the inductive hypothesis, we have

$$\mathrm{tr}(AP_{\mathcal{H}'}) = \mathrm{tr}(A'P_{\mathcal{H}'}) \leq \sum_{i \in [m-1]} \lambda_i(A') = \sum_{i=2}^{m} \lambda_i(A),$$

and, consequently,

$$\mathrm{tr}(AP_{\mathcal{H}}) = \sum_{i \in [r]} \langle g_i, Ag_i \rangle = \mathrm{tr}(AP_{\mathcal{H}'}) + \langle g_m, Ag_m \rangle \leq \sum_{i=2}^{m} \lambda_i(A) + \lambda_1(A) = \sum_{i \in [m]} \lambda_i(A),$$

which completes the proof. $\qquad \square$

## C    SCORE FUNCTIONAL

We start by extending problem (4) to the representation learning for stable non-stationary processes $(X_t)_{t \in \mathbb{N}_0}$. Since for the operator regression problem we typically collect data samples uniformly at random along trajectory, we have that $X \sim \mu := \frac{1}{n} \sum_{i \in [n]} \mu_{i-1}$, recalling that $\mu_t$ is the law

of $X_t$ and $n$ is the sample size. Then, the one step ahead evolution of a random variable $X$ is $X' \sim \mu' = \frac{1}{n}\sum_{i\in[n]}\mu_i$. Since the transfer operator is properly defined as $\mathcal{T}: L^2_{\mu'}(\mathcal{X}) \to L^2_\mu(\mathcal{X})$, we obtain the following version of (3)

$$\left\|[I - P_\mathcal{H}]\mathcal{T}P'_\mathcal{H}\right\|^2 \lambda^+_{\min}(C'_\mathcal{H}) \leq \left\|[I - P_\mathcal{H}]\mathcal{T}_{|\mathcal{H}}\right\|^2 \leq \left\|[I - P_\mathcal{H}]\mathcal{T}P'_\mathcal{H}\right\|^2 \lambda_{\max}(C'_\mathcal{H}), \quad (21)$$

where, $P_\mathcal{H}: L^2_\mu(\mathcal{X}) \to L^2_\mu(\mathcal{X})$ and $P'_\mathcal{H}: L^2_{\mu'}(\mathcal{X}) \to L^2_{\mu'}(\mathcal{X})$ are the orthogonal projections onto $\mathcal{H}$ as a subspace of $L^2_\mu(\mathcal{X})$ and $L^2_{\mu'}(\mathcal{X})$, respectively, and covariance $C'_\mathcal{H}$ is now w.r.t. measure $\mu'$. Now, assuming that the feature map of $\mathcal{H}$ is bounded by some constant $c_\mathcal{H}$, we have that

$$\left\|C'_\mathcal{H} - C_\mathcal{H}\right\| \leq \int_\mathcal{X} \left\|\psi(x) \otimes \psi(x)\right\| |\mu' - \mu|(dx) \leq 2c^2_\mathcal{H}\left\|\mu' - \mu\right\|_{TV} = \frac{2c^2_\mathcal{H}}{n}\left\|\mu_n - \mu_0\right\|_{TV}.$$

Thus, if the dynamics is stable, the total variation norm $\left\|\mu_n - \mu_0\right\|_{TV}$ is bounded w.r.t. $n \to \infty$, and we conclude that for large enough sample size $n$ one can replace $C'_\mathcal{H}$ by $C_\mathcal{H}$ in (21) to obtain tight approximation of the approximation error $\left\|[I - P_\mathcal{H}]\mathcal{T}_{|\mathcal{H}}\right\|^2$ by controlling $\left\|[I - P_\mathcal{H}]\mathcal{T}P'_\mathcal{H}\right\|$. This motivates the optimization problem for non-stationary dynamics

$$\max_{\mathcal{H}\subset L^2_\mu(\mathcal{X}), \mathcal{H}'\subset L^2_{\mu'}(\mathcal{X})} \left\{\left\|P_\mathcal{H}\mathcal{T}P'_{\mathcal{H}'}\right\|^2_{\mathrm{HS}} \mid C_\mathcal{H} = C'_{\mathcal{H}'} = I, \dim(\mathcal{H}) \leq r, \dim(\mathcal{H}') \leq r\right\}. \quad (22)$$

Next, we prove (7), relating the projection score with the covariance operators.

**Lemma 3.** *Let $X$ and $X'$ be two $\mathcal{X}$-valued random variables distributed according to probability measures $\mu$ and $\mu'$, respectively. Given $w \in \mathcal{W}$ let for every $j \in [r]$ functions $\psi_{w,j}: \mathcal{X} \to \mathbb{R}$ and $\psi'_{w,j}: \mathcal{X} \to \mathbb{R}$ be square integrable w.r.t measures $\mu$ and $\mu'$, respectively. If $P_{\mathcal{H}_w}: L^2_\mu(\mathcal{X}) \to L^2_\mu(\mathcal{X})$ and $P'_{\mathcal{H}'_w}: L^2_{\mu'}(\mathcal{X}) \to L^2_{\mu'}(\mathcal{X})$ are orthogonal projections onto subspaces $\mathcal{H}_w := \mathrm{span}(\psi_{w,j})_{j\in[r]}$ and $\mathcal{H}'_w := \mathrm{span}(\psi'_{w,j})_{j\in[r]}$, respectively, then*

$$\left\|P_{\mathcal{H}_w}\mathcal{T}P'_{\mathcal{H}'_w}\right\|^2_{\mathrm{HS}} = \left\|(C^w_X)^{\dagger/2}C^w_{XX'}(C^w_{X'})^{\dagger/2}\right\|^2_{\mathrm{HS}}. \quad (23)$$

*Proof.* The proof directly follows from the notion of finite-dimensional RKHS. Let $k^w_X: \mathcal{X}\times\mathcal{X} \to \mathbb{R}$ and $k^w_{X'}: \mathcal{X}\times\mathcal{X} \to \mathbb{R}$ be two kernels given by

$$k^w_X(x,y) := \psi_w(x)^\top\psi_w(y) \quad\text{and}\quad k^w_{X'}(x,y) := \psi'_w(x)^\top\psi'_w(y),\ x,y \in \mathcal{X}.$$

Then $\mathcal{H}_w$ and $\mathcal{H}'_w$ are the reproducing kernel Hilbert spaces (RKHS) associated with kernels $k^w_X$ and $k^w_{X'}$, respectively.

Now, due to square integrability of the embeddings $\psi_{w,j}$ and $\psi'_{w,j}$, $j \in [r]$, we have that the injection operators of the two RKHS spaces into their respective $L^2$ spaces are well-defined: $S_\mu: \mathcal{H}_w \hookrightarrow L^2_\mu(\mathcal{X})$ and $S_{\mu'}: \mathcal{H}'_w \hookrightarrow L^2_{\mu'}(\mathcal{X})$. Moreover, observing that $\mathcal{H}_w$ and $\mathcal{H}'_w$ are isometrically isomorphic to $\mathbb{R}^r$, we have that $S^*_\mu S_\mu: \mathcal{H}_w \to \mathcal{H}_w$ and $S^*_{\mu'}S_{\mu'}: \mathcal{H}'_w \to \mathcal{H}'_w$ can be identified with $C^w_X \in \mathbb{R}^{r\times r}$ and $C^w_{X'} \in \mathbb{R}^{r\times r}$, respectively, that is

$$S^*_\mu S_\mu = QC^w_X Q^* \quad\text{and}\quad S^*_{\mu'}S_{\mu'} = Q'C^w_{X'}(Q')^*.$$

where $Q: \mathbb{R}^r \to \mathcal{H}_w$ and $Q': \mathbb{R}^r \to \mathcal{H}'_w$ are partial isometries, meaning that $QQ^*$ and $Q'(Q')^*$ are identity operators on $\mathcal{H}_w$ and $Q^*Q$ and $(Q')^*Q'$ are identity matrices in $\mathbb{R}^{r\times r}$.

As a consequence, the polar decompositions of finite rank operators $S_\mu$ and $S_{\mu'}$ can be written as

$$S_\mu = UQ(C^w_X)^{1/2}Q^* \quad\text{and}\quad S_{\mu'} = U'Q'(C^w_{X'})^{1/2}(Q')^*, \quad (24)$$

where $U: \mathcal{H}_w \to L^2_\mu(\mathcal{X})$ and $U': \mathcal{H}'_w \to L^2_{\mu'}(\mathcal{X})$ are partial isometries.

But then, since $\mathcal{H}_w$ as a subspace of $L^2_\mu(\mathcal{X})$ is identified as $\mathrm{Im}(S_\mu)$, and $\mathcal{H}'_w$ as a subspace of $L^2_{\mu'}(\mathcal{X})$ is identified as $\mathrm{Im}(S_{\mu'})$, using adjoints $S^*_\mu: L^2_\mu(\mathcal{X}) \to \mathcal{H}_w$ and $S^*_{\mu'}: L^2_{\mu'}(\mathcal{X}) \to \mathcal{H}'_w$, we can write the aforementioned orthogonal projections as

$$P_{\mathcal{H}_w} = S_\mu(S^*_\mu S_\mu)^\dagger S^*_\mu \quad\text{and}\quad P_{\mathcal{H}'_w} = S_{\mu'}(S^*_{\mu'}S_{\mu'})^\dagger S^*_{\mu'}, \quad (25)$$

which, using (24) and the fact that $(C_X^w)^{1/2}(C_X^w)^\dagger = (C_X^w)^{\dagger/2}$, is equivalent to

$$P_{\mathcal{H}_w} = UQ(C_X^w)^{\dagger/2}Q^*S_\mu^* \quad \text{and} \quad P_{\mathcal{H}'_w} = U'Q'(C_{X'}^w)^{\dagger/2}(Q')^*S_{\mu'}^*.$$

Finally, since for every $v \in \mathbb{R}^r$ we obtain

$$(Q^*S_\mu^*\mathcal{T}S_{\mu'}Q')v = Q^*S_\mu^*(\mathcal{T}S_{\mu'}Q'v) = Q^* \int_{\mathcal{X}} \mu(dx)\phi_X^w(x)(\mathcal{T}S_{\mu'}Q'v)(x)$$

$$= Q^* \int_{\mathcal{X}} \int_{\mathcal{X}} \mu(dx)p(x,dx')\phi_X^w(x)(Q'v)(x')$$

$$= Q^* \int_{\mathcal{X}\times\mathcal{X}} \rho(dx,dx')\phi_X^w(x)(Q'v)(x')$$

where $\rho$ is joint measure of $(X, X')$ and $\phi_X^w$ is the canonical feature map of $k_X^w$, that is $\phi_X^w(x) = k_X^w(\cdot, x)$, $x \in \mathcal{X}$. Next, using the reproducing property

$$(Q^*S_\mu^*\mathcal{T}S_{\mu'}Q')v = Q^* \int_{\mathcal{X}\times\mathcal{X}} \rho(dx,dx')\phi_X^w(x)\langle\phi_{X'}^w(x'), Q'v\rangle_{\mathcal{H}'_w}$$

$$= \left[\iint_{\mathcal{X}\times\mathcal{X}} \rho(dx,dx')Q^*\phi_X^w(x) \otimes ((Q')^*\phi_{X'}^w(x'))\right]v$$

$$= \left[\int_{\mathcal{X}\times\mathcal{X}} \rho(dx,dx')\psi_w(x) \otimes \psi'_w(x')\right]v = C_{XX'}^w v,$$

where $\phi_{X'}^w$ is the canonical feature map of $k_{X'}^w$ and we used that $Q^*\phi_X^w(x) = \psi_w(x)$ and $(Q')^*\phi_{X'}^w(x') = \psi'_w(x')$.

Therefore, using (25) we obtain

$$P_{\mathcal{H}_w}\mathcal{T}P'_{\mathcal{H}'_w} = UQ(C_X^w)^{\dagger/2}C_{XX'}^w(C_{X'}^w)^{\dagger/2}(Q')^*(U')^*,$$

which using that $U, U'$ and $Q, Q'$ are partial isometries, implies (23). $\qquad\square$

Now, using Lem. 1, we prove one of the main theoretical results of the paper. From this point forward, we abbreviate $\mathcal{P}(w) := \mathcal{P}^0(w)$, $\mathcal{S}(w) := \mathcal{S}^0(w)$, $\widehat{\mathcal{P}}_n(w) := \mathcal{P}_n^0(w)$ and $\widehat{\mathcal{S}}_n(w) := \mathcal{P}_n^0(w)$.

**Theorem 1.** *If $\mathcal{T}: L_{\mu'}^2(\mathcal{X}) \to L_\mu^2(\mathcal{X})$ is compact, $\mathcal{H}_w \subseteq L_\mu^2(\mathcal{X})$ and $\mathcal{H}'_w \subseteq L_{\mu'}^2(\mathcal{X})$, then for all $\gamma \geq 0$*

$$\mathcal{S}^\gamma(w) \leq \mathcal{P}^\gamma(w) \leq \sigma_1^2(\mathcal{T}) + \cdots + \sigma_r^2(\mathcal{T}). \tag{10}$$

*Moreover, if $(\psi_{w,j})_{j\in[r]}$ and $(\psi'_{w,j})_{j\in[r]}$ are the leading $r$ left and right singular functions of $\mathcal{T}$, respectively, then both equalities in (10) hold. Finally, if the operator $\mathcal{T}$ is Hilbert-Schmidt, $\sigma_r(\mathcal{T}) > \sigma_{r+1}(\mathcal{T})$ and $\gamma > 0$, then the "only if" relation is satisfied up to unitary equivalence.*

*Proof.* Recall that $\mathcal{P}^\gamma(w) = \mathcal{P}(w) - \gamma(\mathcal{R}(C_X^w) + \mathcal{R}(C_{X'}^w))$ with $\mathcal{P}(w) = \left\|P_{\mathcal{H}_w}\mathcal{T}P'_{\mathcal{H}'_w}\right\|_{\mathrm{HS}}^2 = \left\|(C_X^w)^{\dagger/2}C_{XX'}^w(C_{X'}^w)^{\dagger/2}\right\|_{\mathrm{HS}}^2$, where the last equality follows from Lemma 3, and that $\mathcal{S}(w) := \left\|C_{XX'}^w\right\|_{\mathrm{HS}}^2/(\|C_X^w\|\|C_{X'}^w\|)$. The first inequality in (10) holds thanks to a standard matrix norm inequality, while the second holds by applying Lemma 1(i) and noting that $\mathcal{R}(C_X^w) + \mathcal{R}(C_{X'}^w) \geq 0$.

Now assume that $(\psi_{w,j})_{j\in[r]}$ and $(\psi'_{w,j})_{j\in[r]}$ are some leading $r$ left and right singular functions of $\mathcal{T}$, respectively. Then, since singular functions form ortho-normal systems in $L^2$ spaces we have that

$$(C_X^w)_{i,j} = \langle\psi_{w,i}, \psi_{w,j}\rangle_{L_\mu^2(\mathcal{X})} = \delta_{i,j} \text{ and } (C_{X'}^w)_{i,j} = \langle\psi'_{w,i}, \psi'_{w,j}\rangle_{L_\mu^2(\mathcal{X})} = \delta_{i,j}, i,j \in [r],$$

that is $C_X^w = C_{X'}^w = I_r$, and, therefore, $\mathcal{R}(C_X^w) = \mathcal{R}(C_{X'}^w) = 0$ and $\mathcal{S}(w) = \mathcal{P}(w)$. Thus, using Lemma 1(ii), we obtain that (10) holds with equalities in place of inequalities.

Next, assume that the operator $\mathcal{T}$ is Hilbert-Schmidt, $\sigma_r(\mathcal{T}) > \sigma_{r+1}(\mathcal{T})$, $\gamma > 0$ and $\mathcal{S}^\gamma(w) = \sigma_1^2(\mathcal{T}) + \ldots + \sigma_r^2(\mathcal{T})$. Then, clearly $\mathcal{R}(C_X^w) = \mathcal{R}(C_{X'}^w) = 0$, which implies that $C_X^w = C_{X'}^w = I_r$, that is $(\psi_{w,j})_{j\in[r]}$ and $(\psi'_{w,j})_{j\in[r]}$ form orthonormal systems. Consequently,

$$P_{\mathcal{H}_w} = \sum_{j\in[r]} \psi_{w,j} \otimes \psi_{w,j} \quad \text{and} \quad P_{\mathcal{H}'_w} = \sum_{j\in[r]} \psi'_{w,j} \otimes \psi'_{w,j},$$

and

$$\mathcal{P}(w) = \mathcal{P}^\gamma(w) = \mathcal{S}^\gamma(w) = \sigma_1^2(\mathcal{T}) + \ldots + \sigma_r^2(\mathcal{T}).$$

But then, Lemma 1(iii) implies that $P_{\mathcal{H}_w}$ and $P_{\mathcal{H}'_w}$ are orthogonal projectors on the leading $r$ left and right singular spaces of $\mathcal{T}$. In other words, up to unitary changes of basis, $(\psi_{w,j})_{j \in [r]}$ and $(\psi'_{w,j})_{j \in [r]}$ are the leading $r$ left and right singular functions of $\mathcal{T}$, which completes the proof. $\qquad \square$

We conclude this section with a remark on the importance of regularization.

**Remark 2.** *Recalling the proof of the previous theorem, note that when $\gamma = 0$, equality in (10) is achieved whenever $\mathcal{H}_w$ and $\mathcal{H}'_w$ are spanned by leading $r$ left and right singular functions of $\mathcal{T}$, respectively. Meaning that after the change of basis $Q, Q' \in \mathbb{R}^{r \times r}$ so that $(\sum_{i \in [r]} Q_{i,j} \psi_{w,i})_{j \in [r]}$ and $(\sum_{i \in [r]} Q'_{i,j} \psi'_{w,i})_{j \in [r]}$ are the leading left and right singular functions, respectively. Under the additional conditions, also the "only if" part holds for some changes of basis. Indeed, we can take the change of basis to be $Q = (C_X^w)^{-1/2}$ and $Q' = (C_{X'}^w)^{-1/2}$, which without regularization term, need not be unitary. This, as we see in App. D.2, highly impacts on the stability of computation of the transfer operator estimators, and, therefore, impacts their practical use.*

# D  METHODS

## D.1  STATISTICAL LEARNING GUARANTEES

Before we study the statistical learning guarantees for our novel score $\mathcal{S}$, we first discuss a fundamental limitation of using the score $\mathcal{P}$ to learn the invariant representation. In order to maximize the score $\mathcal{P}$ one can use standard ridge regularization on the empirical covariances. This approach considered in DeepCCA (Andrew et al., 2013) and VAMPNets (Mardt et al., 2019), typically requires a large number of training samples $n$, and the rates are governed by the choice of the regularization hyperparameter, typically ranging from $O(n^{-1/3})$ to $O(n^{-1/2})$. Namely, in this approach one uses score $\|(\widehat{C}_X^w + \lambda I)^{-1/2} \widehat{C}_{XX'}^w (\widehat{C}_{X'}^w + \lambda I)^{-1/2}\|_{\mathrm{HS}}$ instead of $\mathcal{P}$, where $\lambda$ is the regularization parameter. So the main concern is now to measure how close the regularized empirical score is to the true score? To do this, we first study the operator norm deviation:

$$\left| \|(C_X^w)^{\dagger/2} C_{XX'}^w (C_{X'}^w)^{\dagger/2}\| - \|(\widehat{C}_X^w + \lambda I)^{-1/2} \widehat{C}_{XX'}^w (\widehat{C}_{X'}^w + \lambda I)^{-1/2}\| \right|$$

$$\leq \left\| (C_X^w)^{\dagger/2} C_{XX'}^w (C_{X'}^w)^{\dagger/2} - (\widehat{C}_X^w + \lambda I)^{-1/2} \widehat{C}_{XX'}^w (\widehat{C}_{X'}^w + \lambda I)^{-1/2} \right\|$$

$$\leq \left\| (C_X^w)^{\dagger/2} C_{XX'}^w (C_{X'}^w)^{\dagger/2} - (C_X^w + \lambda I)^{-1/2} C_{XX'}^w (C_{X'}^w + \lambda I)^{-1/2} \right\|$$

$$\quad + \left\| (C_X^w + \lambda I)^{-1/2} C_{XX'}^w (C_{X'}^w + \lambda I)^{-1/2} - (\widehat{C}_X^w + \lambda I)^{-1/2} \widehat{C}_{XX'}^w (\widehat{C}_{X'}^w + \lambda I)^{-1/2} \right\|.$$

Using Lemma 4 in (Ullah and Arora, 2023), we get the following control on the "bias term" of the previous display:

$$\left\| (C_X^w)^{\dagger/2} C_{XX'}^w (C_{X'}^w)^{\dagger/2} - (C_X^w + \lambda I)^{-1/2} C_{XX'}^w (C_{X'}^w + \lambda I)^{-1/2} \right\| \leq 8\sqrt{\lambda}.$$

Next, using Lemma 6 in (Fukumizu et al., 2007) gives the following asymptotic rate of convergence on the "variance" part. Assume that $\lambda = \lambda_n \to 0$ as $n \to \infty$, then

$$\left\| (C_X^w + \lambda I)^{-1/2} C_{XX'}^w (C_{X'}^w + \lambda I)^{-1/2} - (\widehat{C}_X^w + \lambda I)^{-1/2} \widehat{C}_{XX'}^w (\widehat{C}_{X'}^w + \lambda I)^{-1/2} \right\| = O_{\mathbb{P}} \left( \frac{1}{\sqrt{\lambda^3 n}} \right).$$

Combining the last three displays and for the optimal choice $\lambda = \lambda_n = n^{-1/4}$, we obtain

$$\left| \|(C_X^w)^{\dagger/2} C_{XX'}^w (C_{X'}^w)^{\dagger/2}\| - \|(\widehat{C}_X^w + \lambda I)^{-1/2} \widehat{C}_{XX'}^w (\widehat{C}_{X'}^w + \lambda I)^{-1/2}\| \right| = O_{\mathbb{P}}(n^{-1/3}).$$

According again to (Ullah and Arora, 2023), in the most favorable scenario of finite-dimensional spaces with well-conditioned covariance matrices, we can obtain an improved control on the "variance" part of the order of magnitude $O_{\mathbb{P}}(\lambda^{-1/2} n^{-1/2})$. Consequently, taking $\lambda = \lambda_n \asymp n^{-1/2} \to 0$ as $n \to \infty$, the estimation error improves from $O_{\mathbb{P}}(n^{-1/3})$ to $O_{\mathbb{P}}(n^{-1/2})$ in the best-case scenario. While these guarantees were obtained for the spectral norm, one can directly deduce such guarantees also for the HS-norm when the latent space is low-dimensional. This analysis highlights a fundamental limitation of using the score $\mathcal{P}(w)$. On the one hand, the Ridge regularization parameter $\lambda$

cannot be set too small to maintain the numerical stability of the method. On the other hand, the statistical analysis requires that $\lambda = \lambda_n$ converges sufficiently fast to 0 as $n \to \infty$ to guarantees that the empirical score $\widehat{\mathcal{P}}$ approximate the true objective $\mathcal{P}$. These two constraints are antagonistic and it is not clear that they can be simultaneously satisfied in practice. Hence, balancing between numerical stability and convergence to the true objective presents a challenging trade-off when using $\mathcal{P}$ for learning invariant representations. By contrast, our score $\mathcal{S}$ and its relaxed version do not suffer from this limitation. Namely, there is no need to sacrifice numerical stability for statistical accuracy or vice versa since $|\mathcal{S}(w) - \widehat{\mathcal{S}}(w)| = O_{\mathbb{P}}(n^{-1/2})$ in all situations, as we prove it in Theorem 3 below.

Our goal is to derive concentration guarantees for the empirical score $\widehat{\mathcal{S}}$ from the true score $\mathcal{S}$. We focus on time-homogeneous Markovian dynamical systems in the stationary regime with invariant measure $\pi$ which was proposed in (Lasota and Mackey, 1994; Kostic et al., 2022).

Recall the definition of the true and empirical scores

$$\mathcal{S}(w) := \frac{\left\|C_{XX'}^w\right\|_{\mathrm{HS}}^2}{\left\|C_X^w\right\|\left\|C_{X'}^w\right\|} \quad \text{and} \quad \widehat{\mathcal{S}}(w) := \frac{\left\|\widehat{C}_{XX'}^w\right\|_{\mathrm{HS}}^2}{\left\|\widehat{C}_X^w\right\|\left\|\widehat{C}_{X'}^w\right\|}, \tag{26}$$

where, as before,

$$\widehat{C}_X^w := \tfrac{1}{n}\sum_{i\in[n]}\psi_w(x_i)\psi_w(x_i)^\top, \widehat{C}_{X'}^w := \tfrac{1}{n}\sum_{i\in[n]}\psi_w'(x_i')\psi_w'(x_i')^\top \text{ and } \widehat{C}_{XX'}^w := \tfrac{1}{n}\sum_{i\in[n]}\psi_w(x_i)\psi_w'(x_i')^\top.$$

Denote by $\rho$ the joint distribution of $(X, X')$.

We assume that the embeddings are bounded almost surely, that is there exists an absolute constant $c$ such that

$$\operatorname*{ess\,sup}_{x\sim\mu}\|\psi_w(x)\|^2 \le c, \qquad \operatorname*{ess\,sup}_{x'\sim\mu'}\|\psi_w'(x')\|^2 \le c. \tag{27}$$

For any fixed $w$ and any fixed $\delta \in (0,1)$, we assume that $n$ is large enough such that

$$\frac{4c}{3\left\|C_X^w\right\|n}\log\left(12r\delta^{-1}\right) + \sqrt{\frac{2}{\left\|C_X^w\right\|n}\log(12r\delta^{-1})} < \frac{1}{3}. \tag{28}$$

Define

$$\varepsilon_n(\delta) = \frac{4c}{3\left\|C_X^w\right\|n}\log\left(12r\delta^{-1}\right) + \sqrt{\frac{2}{\left\|C_X^w\right\|n}\log(12r\delta^{-1})},$$

and

$$\varepsilon_n''(\delta) := c^2\sqrt{\frac{5\log(18\delta^{-1})}{n}} + c^2\frac{C}{n},$$

where $C$ is some absolute constant.

**Theorem 3.** *Let Conditions (27) and (28) be satisfied. Then we get with probability at least $1 - \delta$*

$$\left|\mathcal{S}(w) - \widehat{\mathcal{S}}(w)\right| \le \mathcal{S}(w)\frac{3\varepsilon_n(\delta)}{1 - 3\varepsilon_n(\delta)} + \frac{\varepsilon_n''(\delta)}{\left\|C_X^w\right\|\left\|C_{X'}^w\right\|(1 - 3\varepsilon_n(\delta))}. \tag{29}$$

*Proof.* By definition of $\mathcal{S}(w)$ and $\widehat{\mathcal{S}}(w)$, we have

$$\left|\widehat{\mathcal{S}}(w) - \mathcal{S}(w)\right| \le \widehat{\mathcal{S}}(w)\frac{\left|\left\|\widehat{C}_X^w\right\|\left\|\widehat{C}_{X'}^w\right\| - \left\|C_X^w\right\|\left\|C_{X'}^w\right\|\right|}{\left\|C_X^w\right\|\left\|C_{X'}^w\right\|} + \frac{\left|\left\|\widehat{C}_{XX'}^w\right\|_{\mathrm{HS}}^2 - \left\|C_{XX'}^w\right\|_{\mathrm{HS}}^2\right|}{\left\|C_X^w\right\|\left\|C_{X'}^w\right\|}$$

$$\le \left|\widehat{\mathcal{S}}(w) - \mathcal{S}(w)\right|\frac{\left|\left\|\widehat{C}_X^w\right\|\left\|\widehat{C}_{X'}^w\right\| - \left\|C_X^w\right\|\left\|C_{X'}^w\right\|\right|}{\left\|C_X^w\right\|\left\|C_{X'}^w\right\|} + \mathcal{S}(w)\frac{\left|\left\|\widehat{C}_X^w\right\|\left\|\widehat{C}_{X'}^w\right\| - \left\|C_X^w\right\|\left\|C_{X'}^w\right\|\right|}{\left\|C_X^w\right\|\left\|C_{X'}^w\right\|}$$

$$+ \frac{\left|\left\|\widehat{C}_{XX'}^w\right\|_{\mathrm{HS}}^2 - \left\|C_{XX'}^w\right\|_{\mathrm{HS}}^2\right|}{\left\|C_X^w\right\|\left\|C_{X'}^w\right\|}.$$

Using (34), (36) below, we get with probability at least $1 - 2\delta$,

$$\frac{\left| \|\widehat{C}_X^w\| \|\widehat{C}_{X'}^w\| - \|C_X^w\| \|C_{X'}^w\| \right|}{\|C_X^w\| \|C_{X'}^w\|} \leq \varepsilon_n(\delta) + \varepsilon_n'(\delta) + \varepsilon_n(\delta)\varepsilon_n'(\delta).$$

Next, (37) gives with probability at least $1 - \delta$

$$\frac{\left| \|\widehat{C}_{XX'}^w\|_{\mathrm{HS}}^2 - \|C_{XX'}^w\|_{\mathrm{HS}}^2 \right|}{\|C_X^w\| \|C_{X'}^w\|} \leq \frac{\varepsilon_n''(\delta)}{\|C_X^w\| \|C_{X'}^w\|}.$$

Under Conditions 27 and 28, we get

$$\varepsilon_n(\delta) = \varepsilon_n'(\delta) = \frac{4c}{3\|C_X^w\| n} \log\left(4r\delta^{-1}\right) + \sqrt{\frac{2}{\|C_X^w\| n} \log(4r\delta^{-1})} < \frac{1}{3},$$

and

$$\varepsilon_n''(\delta) := c^2 \sqrt{\frac{5\log(6\delta^{-1})}{n}} + c^2\frac{C}{n}.$$

An union bound gives with probability at least $1 - 3\delta$,

$$\left|\widehat{\mathcal{S}}(w) - \mathcal{S}(w)\right| \leq \mathcal{S}(w)\frac{3\varepsilon_n(\delta)}{1 - 3\varepsilon_n(\delta)} + \frac{\varepsilon_n''(\delta)}{\|C_X^w\| \|C_{X'}^w\|(1 - 3\varepsilon_n(\delta))}. \tag{30}$$

Replacing $\delta$ with $\delta/3$, we get the result with probability $1 - \delta$. $\qquad\square$

To control the operator norm deviation of the empirical covariances $\widehat{C}_X^w$ and $\widehat{C}_{X'}^w$ from their population counterparts, we use the following dimension-free version of (Minsker, 2017) of the non-commutative Bernstein inequality (see also Theorem 7.3.1 in (Tropp, 2012) for an easier to read and slightly improved version) as well as an extension to self-adjoint Hilbert-Schmidt operators on separable Hilbert spaces.

**Proposition 1** ((Minsker, 2017) and Theorem 7.3.1 in (Tropp, 2012)). *Let $A_i$, $i \in [n]$ be i.i.d copies of a random Hilbert-Schmidt operator $A$ on separable Hilbert spaces. Let $\|A\| \leq c$ almost surely, $\mathbb{E}A = 0$ and let $\mathbb{E}[A^2] \preceq V$ for some trace class operator $V$. Then with probability at least $1 - \delta$*

$$\left\| \frac{1}{n}\sum_{i\in[n]} A_i \right\| \leq \frac{2c}{3n}\mathcal{L}_A(\delta) + \sqrt{\frac{2\|V\|}{n}\mathcal{L}_A(\delta)}, \tag{31}$$

*where*

$$\mathcal{L}_A(\delta) := \log\frac{4}{\delta} + \log\frac{\mathrm{tr}(V)}{\|V\|}.$$

**Proposition 2.** *Assume that $c_\psi := \sup_x \left\{ \|\psi_w(x)\|^2 \right\} < \infty$. Given $\delta > 0$, with probability in the i.i.d. draw of $(x_i)_{i=1}^n$ from $\mu$, it holds that*

$$\mathbb{P}\{\|\widehat{C}_X^w - C_X^w\|/\|C_X^w\| \leq \varepsilon_n(\delta)\} \geq 1 - \delta, \tag{32}$$

*where*

$$\varepsilon_n(\delta) := \frac{4c_\psi}{3\|C_X^w\| n}\mathcal{L}(\delta) + \sqrt{\frac{2}{\|C_X^w\| n}\mathcal{L}(\delta)} \quad \text{and} \quad \mathcal{L}(\delta) := \log\frac{4r}{\delta}. \tag{33}$$

*Proof of Proposition 2.* Proof follows directly from Proposition 1 applied to operators $\psi_w(x_i) \otimes \psi_w(x_i)$ using the fact that $C_X^w = \mathbb{E}\,\psi_w(x_i) \otimes \psi_w(x_i)$, where we recall that $\psi_w(x) := (\psi_{w,1}(x), \ldots, \psi_{w,r}(x)) \in \mathbb{R}^r$. Hence, we have the obvious upper bound $\frac{\mathrm{tr}(C_X^w)}{\|C_X^w\|} \leq r$. $\qquad\square$

We deduce from (32), with probability at least $1 - \delta$

$$(1 - \varepsilon_n(\delta))\|C_X^w\| \le \|\widehat{C}_X^w\| \le \|C_X^w\|(1 + \varepsilon_n(\delta))\|C_X^w\|, \tag{34}$$

A similar result is also valid for $\widehat{C}_{X'}^w$ provided that $c_{\psi'} := \sup_{w,x}\left\{\left\|\psi_w'(x)\right\|^2\right\} < \infty$. Define

$$\varepsilon_n'(\delta) := \frac{4c_{\psi'}}{3\|C_X^w\| n}\mathcal{L}(\delta) + \sqrt{\frac{2}{\|C_X^w\| n}\mathcal{L}(\delta)} \quad \text{and} \quad \mathcal{L}(\delta) := \log\frac{4r}{\delta}. \tag{35}$$

Then, with probability at least $1 - \delta$

$$(1 - \varepsilon_n'(\delta))\|C_{X'}^w\| \le \|\widehat{C}_{X'}^w\| \le (1 + \varepsilon_n'(\delta))\|C_{X'}^w\|, \tag{36}$$

We study now the deviation of $\left\|\widehat{C}_{XX'}^w\right\|_{\mathrm{HS}}^2$ from $\left\|C_{XX'}^w\right\|_{\mathrm{HS}}^2$. We can essentially apply Theorem 3 in (Gretton et al., 2005) with kernels $k_w(x, x') = \langle \psi_w(x), \psi_w(x')\rangle$ and $l_w(y, y') = \langle \psi_w'(y), \psi_w'(y')\rangle$. Note that these two kernels are essentially bounded. Indeed we have

$$\underset{x,x'}{\mathrm{ess\,sup}}\, |k_w(x, x')| \le \sup_x \|\psi_w(x)\|^2 \le c_\psi \quad \text{and} \quad \underset{y,y'}{\mathrm{ess\,sup}}\, |l_w(y, y')| \le \sup_x \|\psi_w'(y)\|^2 \le c_{\psi'}.$$

Hence, for any $n \ge 2$ and $\delta > 0$, we get with probability at least $1 - \delta$,

$$\left| \left\|\widehat{C}_{XX'}^w\right\|_{\mathrm{HS}}^2 - \left\|C_{XX'}^w\right\|_{\mathrm{HS}}^2 \right| \le \epsilon_n''(\delta), \tag{37}$$

where

$$\epsilon_n''(\delta) := c_\psi c_{\psi'}\sqrt{\frac{5\log(6\delta^{-1})}{n}} + c_\psi c_{\psi'}\frac{C}{n},$$

for some absolute constant $C > 0$.

## D.2 OPERATOR REGRESSION AND PREDICTION

We next discuss how to design an estimator of the transfer operator $\mathcal{T}\colon L_{\mu'}^2(\mathcal{X}) \to L_\mu^2(\mathcal{X})$ using the learned subspaces $\mathcal{H}_w$ and $\mathcal{H}_w'$. Namely, we estimate $\mathcal{T} \approx \widehat{\mathcal{T}}_w\colon \mathcal{H}_w' \to \mathcal{H}_w$. The purpose of such estimation is to, given a initial state $x \in \mathcal{X}$, predict the average evolution $\mathbb{E}[f(X') \,|\, X = x]$ of an observable $f \in L_{\mu'}^2(\mathcal{X})$. Remark that in the main text, we have just discussed the task when one takes $\mathcal{H}_w' = \mathcal{H}_w$.

In what follows, let us assume that after the training we obtained $w \in \mathcal{W}$ such that $C_X^w$ and $C_{X'}^w$ are invertible, that is that $(\psi_{w,j})_{j\in[r]}$ $(\psi_{w,j}')_{j\in[r]}$ form basis of the spaces $\mathcal{H}_w$ and $\mathcal{H}_w'$, respectively. This means that the operators $E_w,\colon \mathbb{R}^r \mapsto \mathcal{H}_w$ and $E_w'\colon \mathbb{R}^r \mapsto \mathcal{H}_w'$ can be properly defined as partial isometries by $E_w v = \psi_w(\cdot)^\top v$ and $E_w' v = \psi_w'(\cdot)^\top v$. So, every estimator can be written in the form $\widehat{\mathcal{T}}_w = E_w \widehat{T}(E_w')^*$ for some $\widehat{T} \in \mathbb{R}^{r \times r}$.

Different estimators can then computed from data $\mathcal{D}_n := (x_i, x_i')_{i\in[n]}$ (either seen or unseen during training time). To elaborate on this, let us, as usual for kernel methods, define the sampling operators $\widehat{S}\colon \mathcal{H}_w \to \mathbb{R}^n$ and $\widehat{S}'\colon \mathcal{H}_w' \to \mathbb{R}^n$, given by $\widehat{S}h = n^{-\frac{1}{2}}[h(x_1) \ldots h(x_n)]^\top$, $f \in \mathcal{H}_w$, and $\widehat{S}'g = n^{-\frac{1}{2}}[g(x_1') \ldots g(x_n')]^\top$, $g \in \mathcal{H}_w'$. Notice that we can extend the domain of definition of these operators via interpolation to arbitrary functions $\mathcal{X} \to \mathbb{R}$ that can be evaluated on a dataset, respectively. Hence, without possible confusion, when evaluating we can use $\widehat{S}f$ and $\widehat{S}'f'$ even when $f \notin \mathcal{H}_w$ or $f' \notin \mathcal{H}_w'$.

Now, as shown in (Kostic et al., 2022), the empirical estimator $\widehat{\mathcal{T}}_w$ of the transfer operator $\mathcal{T}$ using dataset $\mathcal{D}_n$ can be obtained via operator regression by minimizing the empirical risk

$$\left\|\widehat{S}' - \widehat{S}\widehat{\mathcal{T}}_w\right\|_{\mathrm{HS}}^2 = \left\|\widehat{S}' - \widehat{S}E_w\widehat{T}(E_w')^*\right\|_{\mathrm{HS}}^2 = \left\|\widehat{S}'E_w' - \widehat{S}E_w\widehat{T}\right\|_{\mathrm{HS}}^2$$

where the last equality holds since $E_w'$ is a partial isometry. Therefore, the simple least square (LS) estimator is than obtained as $\widehat{T} := (E_w^*\widehat{S}^*\widehat{S}E_w)^\dagger(E_w^*\widehat{S}^*\widehat{S}'E_w') = (\widehat{C}_X^w)^\dagger\widehat{C}_{XX'}^w$, or, equivalently, as $\widehat{\mathcal{T}}_w = E_w\widehat{T}(E_w')^* = \widehat{S}^*(\widehat{S}\widehat{S}^*)^\dagger\widehat{S}'$

Once the regression is performed, recalling that $X'$ is a $\Delta t = 1$ step ahead evolution of $X$ we can use it to approximate $\mathbb{E}[f(X') \,|\, X = x] \approx (\widehat{\mathcal{T}}_w f)(x)$ for $f\colon \mathcal{X} \to \mathbb{R}$, as the following result shows.

**Proposition 3.** *Let* $\widehat{\mathcal{T}} \colon \mathcal{H}'_w \to \mathcal{H}_w$ *be a LS estimator of* $\mathcal{T}$, *then for every* $x \in \mathcal{X}$ *and* $f \in \mathcal{H}_w$

$$(\widehat{\mathcal{T}}_w f)(x) = \psi_w(x)^\top (\widehat{C}^w_X)^\dagger \widehat{\Psi}_w [f(x'_1) \,|\, \cdots \,|\, f(x'_n)]^\top, \tag{38}$$

*where* $\widehat{\Psi}_w := [\psi_w(x_1) \,|\, \cdots \,|\, \psi_w(x_n)] \in \mathbb{R}^{r \times n}$.

*Proof.* The proof follows directly observing that $\widehat{\Psi}^\top_w = \sqrt{n}[\widehat{S}\psi_{w,1} \,|\, \ldots \,|\, \widehat{S}\psi_{w,r}] = \sqrt{n}\widehat{S}E_w$. Namely, then

$$\widehat{\Psi}_w [f(x'_1) \,|\, \cdots \,|\, f(x'_n)]^\top = (\widehat{S}E_w)^* \widehat{S}' f = E^*_w \widehat{S}^* \widehat{S}' f = E^*_w \widehat{S}^* \widehat{S}' E_w E^*_w f = \widehat{C}^w_{XX'} E^*_w f$$

and, hence

$$\psi_w(x)^\top (\widehat{C}^w_X)^\dagger \widehat{\Psi}_w [f(x'_1) \,|\, \cdots \,|\, f(x'_n)]^\top = E_w (\widehat{C}^w_X)^\dagger \widehat{C}^w_{XX'} E^*_w f = \widehat{\mathcal{T}}_w f.$$

$\square$

We remark that the previous result formally holds for $f \in \mathcal{H}'_w$, but it can be easily extended to functions in $L^2_{\mu'}(\mathcal{X})$ via interpolation.

### D.3    DYNAMICS MODE DECOMPOSITION AND FORECASTING

Now we consider the problem of forecasting the process for several time steps in future using what is known as (extended) dynamic mode decomposition, which is based on the estimated eigenvalues and eigenfunctions of the transfer operator. As observed in the main body, this is meaningful only if the operator is an endomorphism on a function space, that is if it maps the space into itself.

Hence, after training DPNet we will use just one representation $\psi_w$ and its $r$-dimensional space of functions $\mathcal{H}_w := \mathrm{span}(\psi_{w,j})_{j \in [r]}$ to perform the operator regression, as explained in the previous section, and obtain an estimator $\widehat{\mathcal{T}}_w = E_w \widehat{T} E^*_w \colon \mathcal{H}_w \to \mathcal{H}_w$, for some matrix $\widehat{T} \in \mathbb{R}^{r \times r}$. Then, if $(\widehat{\lambda}_i, \widehat{u}_i, \widehat{v}_i)_{i \in [r]} \subset \mathbb{C} \times \mathbb{C}^r \times \mathbb{C}^r$ is a spectral decomposition of $\widehat{T}$, then $(\widehat{\lambda}_i, E_w \widehat{u}_i, E_w \widehat{v}_i)_{i \in [r]}$ is a spectral decomposition of $\widehat{\mathcal{T}}_w$. In the following result we show how to compute dynamic mode decomposition of $\mathcal{T}$ based on the estimator $\widehat{\mathcal{T}}_w$ and use it for forecasting by approximating $\mathbb{E}[f(X_t) \,|\, X_0 = x] \approx ((\widehat{\mathcal{T}}_w)^t f)(x)$, for $f \colon \mathcal{X} \to \mathbb{R}$, $x \in \mathcal{X}$.

**Proposition 4.** *Let* $\widehat{\mathcal{T}}_w = E_w \widehat{T} E_w \colon \mathcal{H}_w \to \mathcal{H}_w$ *be rank* $r$ *LS estimator of* $\mathcal{T}_{\Delta t}$, *for* $\Delta t = 1$. *If* $\widehat{T} = \sum_{i \in [r]} \widehat{\lambda}_i \widehat{v}_i \widehat{u}^*_i$ *is the spectral decomposition of* $\widehat{T}$, *and* $\widehat{f}_i(x) := \psi_w(x)^\top \widehat{v}_i$ *and* $\widehat{g}_i(x) := (\widehat{u}_i)^* \psi_w(x)$, $i \in [r]$, *then for every* $t \in \mathbb{N}$, *every* $x \in \mathcal{X}$ *and every* $f \in \mathcal{H}_w$ *it holds that*

$$((\widehat{\mathcal{T}}_w)^t f)(x) = \sum_{i \in [r]} \widehat{\lambda}^t_i \, \widehat{f}_i(x) \, \widehat{u}^*_i D_w(f), \tag{39}$$

*where* $D_w(f) := \widehat{\Lambda}^{-1}(\widehat{C}^w_X)^\dagger \widehat{\Psi}_w [f(x'_1) \,|\, \ldots \,|\, f(x'_n)]^\top \in \mathbb{R}^r$.

*Proof.* First observe that

$$(\widehat{\mathcal{T}}_w)^t f = E_w \widehat{T}^t E^*_w f = E_w \widehat{T}^{t-1} E^*_w E_w \widehat{T} E^*_w f = \sum_{i \in [r]} \widehat{\lambda}^{t-1}_i \, (\widehat{f}_i \otimes \widehat{g}_i) E_w \widehat{T} E^*_w f.$$

Hence,

$$((\widehat{\mathcal{T}}_w)^t f)(x) = \sum_{i \in [r]} \widehat{\lambda}^{t-1}_i \, \widehat{f}_i(x)(\widehat{u}^*_i \widehat{T} E^*_w f) = \sum_{i \in [r]} \widehat{\lambda}^t_i \, \widehat{f}_i(x)(\widehat{\lambda}^{-1} \widehat{u}^*_i \widehat{T} E^*_w f)$$

and the rest of the proof follows as in Prop. 3.

$\square$

We remark that $\widehat{u}^*_i D_w(f)$ is known as $i$-th Koopman mode of the observable $f$, and that in comparison to Encoder-Decoder approaches $D_w(f)$ can be considered as a decoder when forecasting function $f \colon \mathcal{X} \to \mathbb{R}$. Clearly this is easily extended to vector valued functions, and, hence, we can forecast the states by using $\widehat{u}^*_i D_w(I) := \widehat{\Lambda}^{-1}(\widehat{C}^w_X)^\dagger \widehat{\Psi}_w [x'_1 \,|\, \ldots \,|\, x'_n]^\top \in \mathbb{R}^d$, where $\mathcal{X} \subset \mathbb{R}^d$.

---

**Algorithm 2** DPNets Training (Discrete)

---

**Input:** data $\mathcal{D}_n = (x_1, \ldots, x_n)$, $\mathcal{D}'_n = (x'_1, \ldots, x'_n)$, metric distortion loss $\mathcal{R}$; optimizer $U$; DNNs
   $\psi_w, \psi'_w \colon \mathcal{X} \to \mathbb{R}^r$; metric loss coefficient $\gamma$; # of steps $K$; minibatch size $m$.
 1: Initialize DNN weights $w_1$
 2: **for** $k = 1$ to $K$ **do**,
 3:     Sample minibatches $(y_1, \ldots, y_m)$ from $\mathcal{D}_n$, and $(y'_1, \ldots, y'_m)$ from $\mathcal{D}'_n$.
 4:     Compute empirical covariance matrices
         $\widehat{C}_X^{w_k} \leftarrow \frac{1}{n} \sum_{i=1}^m \psi_{w_k}(y_i)\psi_{w_k}(y_i)^\top, \widehat{C}_{X'}^{w_k} \leftarrow \frac{1}{n} \sum_{i=1}^m \psi'_{w_k}(y'_i)\psi'_{w_k}(y'_i)^\top$
         $\widehat{C}_{XX'}^{w_k} \leftarrow \frac{1}{n} \sum_{i=1}^m \psi_{w_k}(y_i)\psi'_{w_k}(y'_i)^\top$
 5:     **if** DPNets-relaxed **then**
 6:
         $$F(w_k) \leftarrow \widehat{\mathcal{S}}_m^\gamma(w_k) := \frac{\left\| \widehat{C}_{XX'}^{w_k} \right\|_{\mathrm{HS}}^2}{\left\| \widehat{C}_X^{w_k} \right\| \left\| \widehat{C}_{X'}^{w_k} \right\|} - \gamma\big(\mathcal{R}(\widehat{C}_X^{w_k}) + \mathcal{R}(\widehat{C}_{X'}^{w_k})\big)$$
 7:     **else**
 8:         $F(w_k) \leftarrow \widehat{\mathcal{P}}_m^\gamma(w_k) := \left\| (\widehat{C}_X^{w_k})^{\frac{\dagger}{2}} \widehat{C}_{XX'}^{w_k} (\widehat{C}_{X'}^{w_k})^{\frac{\dagger}{2}} \right\|_{\mathrm{HS}}^2 - \gamma\big(\mathcal{R}(\widehat{C}_X^{w_k}) + \mathcal{R}(\widehat{C}_{X'}^{w_k})\big)$
 9:     **end if**
10:     $w_{k+1} \leftarrow U(w_k, \nabla F(w_k))$ where $\nabla F(w_k)$ is computed via backpropagation
11: **end for**
12: **return** representations $\psi_{w_K}, \psi'_{w_K}$

---

### D.4 Extended Algorithm and Training Time

Algorithm 2 is an extended version of Algorithm 1. The time complexity of computing the empirical scores $\widehat{\mathcal{P}}_m^\gamma(w)$, $\widehat{\mathcal{S}}_m^\gamma(w)$ and (thanks to backpropagation) their gradients, is $O(m\,\mathrm{Cost}(\psi_w) + mr^2 + r^3)$ and $O(m\,\mathrm{Cost}(\psi_w) + mr^2)$ respectively, where $\mathrm{Cost}(\psi_w)$ is the cost of one evaluation of $\psi_w$. Namely, computing the embeddings for $m$ samples costs $O(m\,\mathrm{Cost}(\psi_w))$, computing the (cross)covariance matrices for $m$ samples costs $O(mr^2)$, computing the pseudoinverse via eignevalue decomposition costs $O(r^3)$, while computing the operator norm of the covariance matrices using e.g. the Arnoldi iteration method costs $O(r^2)$. We note that the cost of training VAMPNets (Mardt et al., 2018) is the of the same order as DPNets without relaxation, since evaluating the metric distortion loss is relatively cheap once we have the covariance matrices.

## E SDE learning

We next prove the second main result on the optimization problem (5).

**Theorem 2.** *If $\mathcal{H}_w \subseteq W_\pi^{1,2}(\mathcal{X})$, and $\lambda_1(\mathcal{L}) \geq \cdots \geq \lambda_{r+1}(\mathcal{L})$ are eigenvalues of $\mathcal{L}$ above its essential spectrum, then for every $\gamma \geq 0$ it holds*

$$\mathcal{P}_\partial^\gamma(w) := \mathrm{tr}\left( (C_X^w)^\dagger C_{X\partial}^w \right) - \gamma\mathcal{R}(C_X^w) \leq \lambda_1(\mathcal{L}) + \cdots + \lambda_r(\mathcal{L}), \tag{14}$$

*and the equality is achieved when $\psi_{w,j}$ is the eigenfunction of $\mathcal{L}$ corresponding to $\lambda_j(\mathcal{L})$, for $j \in [r]$.*

*Proof.* In view of Lem. 2, we only need to prove that $\mathrm{tr}(P_{\mathcal{H}_w} \mathcal{L} P_{\mathcal{H}_w}) = \mathrm{tr}\left( (C_X^w)^\dagger C_{X\partial}^w \right)$. To that end, we reason as in the proof of Lem. 3 to obtain that $P_{\mathcal{H}_w} = UQ(C_X^w)^{\dagger/2}Q^* S_\pi^*$, where $S_\pi \colon \mathcal{H}_w \hookrightarrow W_\pi^{1,2}(\mathcal{X})$ is an injection and $Q \colon \mathbb{R}^r \to \mathcal{H}_w$ and $U \colon \mathcal{H}_w \to W_\pi^{1,2}(\mathcal{X})$ are partial isometries. Moreover, recalling (2), we have that

$$\begin{aligned}
(Q^* S_\pi^* \mathcal{L} S_\pi Q) &= \lim_{\Delta t \to 0^+} \frac{Q^* S_\pi^* (\mathcal{T}_{\Delta t} - I) S_\pi Q}{\Delta t} \\
&= \lim_{\Delta t \to 0^+} Q^* \left[ \int_{\mathcal{X}} \pi(dx)\phi_X^w(x) \otimes \frac{\int_{\mathcal{X}} p_{\Delta t}(x, dx')(Q^* \phi_X^w(x') - Q^* \phi_X^w(x))}{\Delta t} \right] \\
&= \int_{\mathcal{X}} \pi(dx)\psi_w(x) \otimes \left( \lim_{\Delta t \to 0^+} \frac{\int_{\mathcal{X}} p_{\Delta t}(x, dx')(\psi_w(x') - \psi_w(x))}{\Delta t} \right) \\
&= \mathbb{E}_{X \sim \pi}\left[ \psi_w(X) \otimes d\psi_w(X) \right] = C_{X\partial}^w,
\end{aligned}$$

|  | DPNets | DPNets-relax | VAMPNets | DynAE | ConsAE |
|---|---|---|---|---|---|
| **Logistic** | $1 \times 10^{-4}$ | $1 \times 10^{-4}$ | $3 \times 10^{-6}$ | - | - |
| **Fluid** | Failed | $1 \times 10^{-4}$ | Failed | $9 \times 10^{-4}$ | $9 \times 10^{-4}$ |
| **MNIST** | $9 \times 10^{-4}$ | $9 \times 10^{-4}$ | $9 \times 10^{-4}$ | $9 \times 10^{-4}$ | $9 \times 10^{-4}$ |
| **Chignolin** | Failed | $1 \times 10^{-3}$ | Failed | - | - |
| **Langevin** | $1 \times 10^{-3}$ | - | - | - | - |

Table 3: Best learning rates found by a grid search for each (experiment, method) pair. For all experiments, the grid was made by 100 equally spaced values in the interval $(10^{-6}, 10^{-2})$.

where the last line follows from Itō formula (see e.g. Arnold, 1974). Hence, using $P_{\mathcal{H}_w} = UQ(C_X^w)^{\dagger/2}Q^*S_\pi^*$ we have that

$$\text{tr}(P_{\mathcal{H}_w}\mathcal{L}P_{\mathcal{H}_w}) = \text{tr}((C_X^w)^{\dagger/2}C_{X\partial}^w(C_X^w)^{\dagger/2}) = \text{tr}((C_X^w)^\dagger C_{X\partial}^w),$$

which completes the proof $\qquad\square$

To conclude this section, we remark that in the continuous setting the estimator $\widehat{\mathcal{L}}_w \colon \mathcal{H}_w \to \mathcal{H}_w$ of $\mathcal{L}$ on the learned space $\mathcal{H}_w$ can be obtained via operator regression in a similar way as discussed in Sec. 4. So, the LS estimator is given by matrix $\widehat{L} = (\widehat{C}_X^w)^\dagger \widehat{C}_{X\partial}^w$, and its spectral decomposition is

$$\widehat{\mathcal{L}}_w = \sum_{i \in [r]} \widehat{\lambda}_i \, \widehat{f}_i \otimes \widehat{g}_i, \quad \text{where} \quad \widehat{f}_i(x) := \psi_w(x)^\top \widehat{v}_i \text{ and } \widehat{g}_i(x) := (\widehat{u}_i)^* \psi_w(x), \quad (40)$$

where $\widehat{L} = \sum_{i \in [r]} \widehat{\lambda}_i \widehat{v}_i \widehat{u}_i^\top$ is the spectral decomposition of the matrix $\widehat{L}$.

Hence, using that $\lambda_i(\mathcal{T}_{\Delta t}) = \exp(\lambda_i(\mathcal{L}))$, we directly obtain the modal decomposition in the continuous time,

$$\mathbb{E}[f(X_t) \,|\, X_0 = x] \approx \sum_{i \in [r]} \exp(\widehat{\lambda}_i \, t) \, \widehat{f}_i(x) \, \widehat{u}_i^* D_w(f), \quad t \in [0, +\infty) \quad (41)$$

where $D_w(f) = (\widehat{C}_X^w)^\dagger \widehat{\Psi}_w[f(x_1)\,|\,\cdots\,|f(x_n)]^\top \in \mathbb{R}^r$ are the coefficients of the LS estimator of $f$ in $\mathcal{H}_w$. As a final remark, note that in (41) we regress function $f$ onto $\mathcal{H}_w$ using least squares with data $(x_i, f(x_i))$.

## F EXPERIMENTS

**Hardware** The experiments were performed on a workstation equipped with an Intel(R) Core™i9-9900X CPU @ 3.50GHz, 48GB of RAM and a NVIDIA GeForce RTX 2080 Ti GPU. Due to RAM insufficiency, the Nyström baseline reported in Table 2 was performed on a CPU node of a cluster with 2x AMD EPYC 7713 @ 2.0GHz and 512GB of RAM.

**Software** All experiments and baselines have been implemented in Python 3.11 and Pytorch 2.0, the only exception being the PINNs baseline for the fluid flow experiment, for which we relied on the original implementation of the code. All the code to reproduce the experiments will be made openly available.

**General remarks** Every algorithm has been performed in `float32` single precision, fixing the random number generator seed where appropriate. We made sure that each algorithm was trained on the same combinations of input-output data, and for neural network models we used the same batches and number of epochs. The learning rate for each method was tuned independently by running a small number of steps at 100 equally spaced learning rates in the interval $(10^{-6}, 10^{-2})$ and selecting the best out of these. The best learning rates for each experiment are reported in Table 3

Once a representation was learned, we always used the Ordinary Least Squares estimator described in 4 to perform the subsequent evaluation tasks.

### F.1 LOGISTIC MAP

**Data** We generated the data as explained in Kostic et al. (2022) for a value $N = 20$ of the trigonometric noise. To train DPNets and VAMPNets we have sampled a trajectory of $2^{14} \approx 16000$ points. We used a batch size of $2^{13}$ points and trained for 500 epochs.

**Optimization** Adam with learning rate tuned as explained in the general remarks.

**Architecture** Multi layer perceptron of shape `Linear[64]`→`Linear[128]`→ `Linear[64]`→`Linear[feature_dim]` with Leaky ReLU activations. We have set the dimension of the feature map to $r = 7$ as the minimal dimension allowing to meaningfully learn the first three eigenvalues.

**Additional results: the role of feature dimension** In the left panel of Fig. 5 we compare the eigenvalues and the singular values of $\mathcal{T}$ for the noisy logistic map, while in the center and right panels we show the metrics reported in Tab. 1 of the main text as a function of the feature dimension $r$. Notice how the singular values decay significantly slower than eigenvalues, a consequence of the fact that the transfer operator is not normal, i.e. $\mathcal{T}\mathcal{T}^* \neq \mathcal{T}^*\mathcal{T}$. Non-normality makes the estimation of the spectra of $\mathcal{T}$ particularly sensitive, as captured by the pseudospectra, see Trefethen and Embree (2020). In Fig. 6 we show how small estimation errors in operator norm (label on the contour lines) incur larger errors in the eigenvalue estimation (distance of the true eigenvalues to the contour lines in the complex plane). This means that for non-normal operators the estimation error needs to be typically much smaller than the modulus of the eigenvalues one wants to recover. For $r < 7$, the spectral error of every model is of the same order of the eigenvalues to approximate, as unequivocally shown in Fig. 6. At $r = 7$, DPNets-relaxed already give a decent approximation of the three leading eigenvalues, see also Fig. 6. From $r > 7$ onward, every model progressively yields reasonable estimations, with DPNets and Cheby-T quickly catching up with DPNets-relaxed. For the optimality gap, every model shows an improving trend by increasing the feature dimension $r$, with DPNets showing the strongest performance.

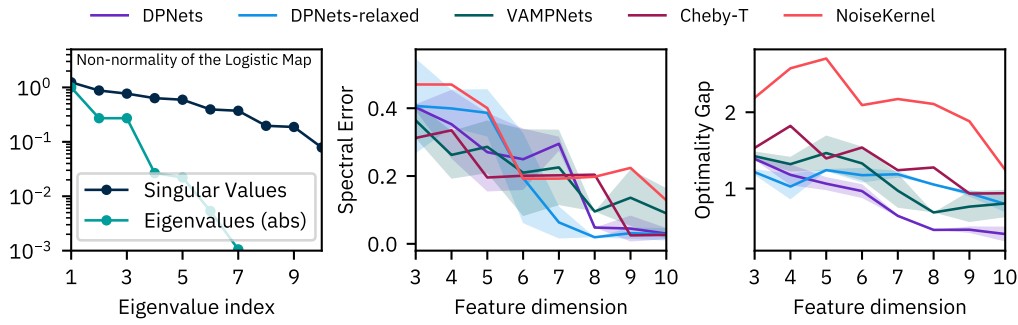

Figure 5: (Left) Decay of the singular values $\sigma_i(\mathcal{T})$ and of the the eigenvalues $|\lambda_i(\mathcal{T})|$ for the logistic map example. (Center, Right) Spectral error and Optimality gap as a function of the feature dimension $r$ for the baselines considered in Tab. 1.

**Hyperparameters** Metric loss coefficient $\gamma = 1$.

The reported results concern the average over 20 independent seed initializations.

## F.2 FLUID FLOW PAST A CYLINDER

The data for this experiment are equally spaced sampled solutions of the Navier-Stokes equations (Trefethen and Embree, 2020) for an incompressible Newtonian fluid coupled with the transport equation

$$\partial_t c + \boldsymbol{u} \cdot \nabla c = \mathrm{Pec}^{-1}\nabla^2 c.$$

Here, Pec is the Péclet number, and $c : \mathbb{R}^3 \to \mathbb{R}$ is a field representing the concentration $c(t; x, y)$ of a scalar quantity which is transported by the fluid flow without influencing the fluid motion itself (e.g. a dye dissolved in water). These partial derivative equations are solved over a $100 \times 200$ regular grid.

In this experiment we have only been able to train DPNets-relaxed. Indeed both DPNets (unrelaxed) and VAMPNets failed due to linear algebra errors arising in the back propagation step for the pseudo-inverse matrix.

**Data** Available at `https://github.com/maziarraissi/HFM`. It consists of 201 snapshots: 160 used for training, the rest for testing. Has been standardized: each channel with its own mean and std. In Fig. 8 we show two snapshots from the training dataset.

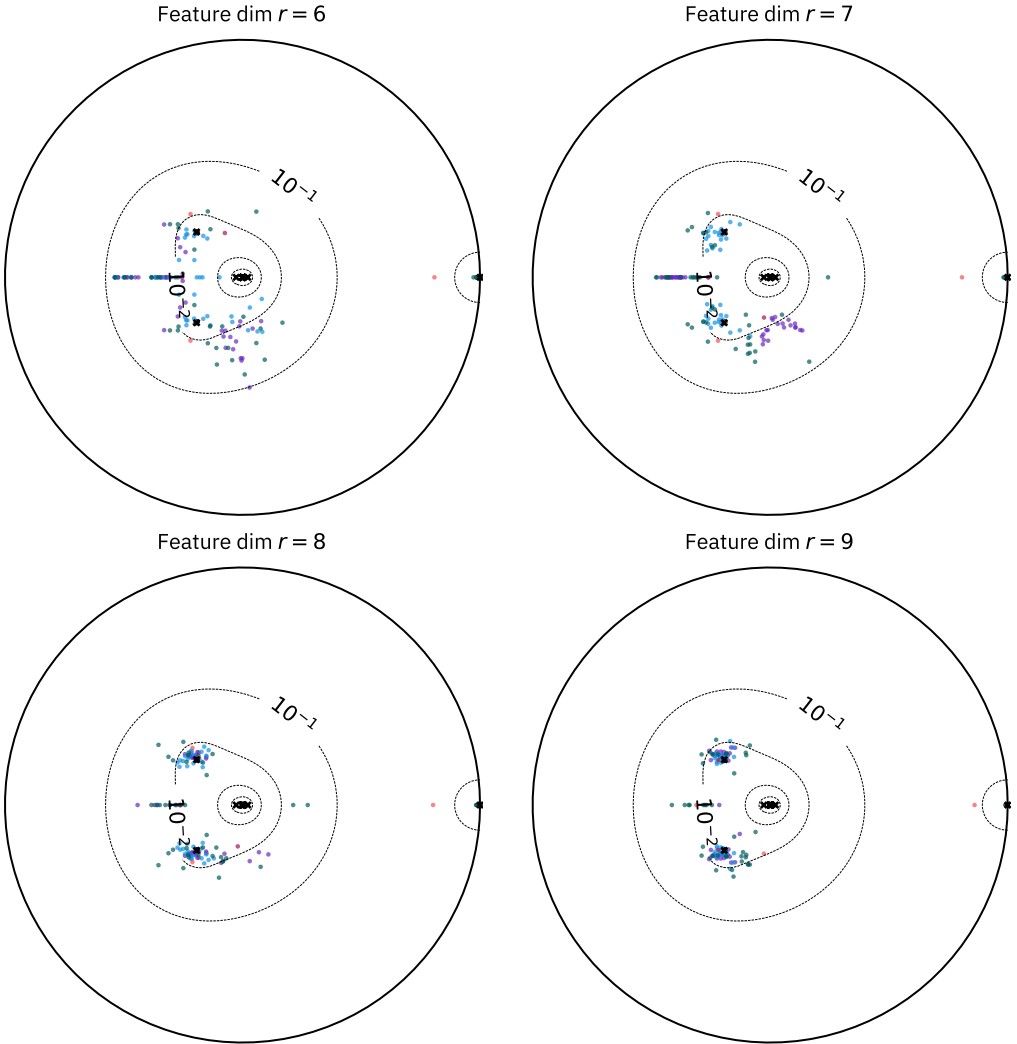

Figure 6: Estimation of the eigenvalues of the logistic map at different feature dimensions. For each model, we plot the eigenvalues estimated by 20 independent initial seeds. The color in each scatter plot follows the same color-coding of Fig. 5. Black $\times$ represent reference eigenvalues, while the dashed contour lines show the pseudospectral regions when the estimation error ranges from $10^{-1}$ (outermost) to $10^{-4}$ (innermost). Note that for this problem pseudospectra indicates that the leading eigenvalue is easy to recover, while recovering eigenvalues close to zero is very hard.

**Optimization** Adam with learning rate tuned as explained in the general remarks. Full-batch training. 5000 total training iterations/epochs. The training time of a full batch $\approx 39$ mins.

**Architecture** MLP with layers of width `[128, 512, 1024, 512, 256, 64]` and ReLU activation function

**Hyperparameters** Metric loss coefficient $\gamma = 1$;

### F.3 CONTINUOUS DYNAMICS

The implementation of this experiment is straightforward, and our results can be reproduced using the following informations.

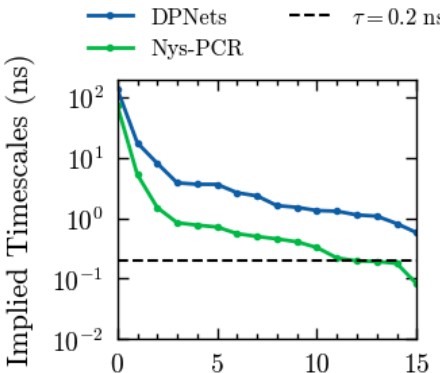

Figure 7: The implied timescale (Mardt et al., 2018) associated to the eignvalues of the transfer operator. Everything which is below the simulation lagtime $\tau = 0.2$ ns, is just numerical noise.

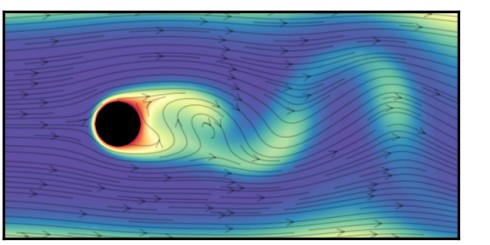 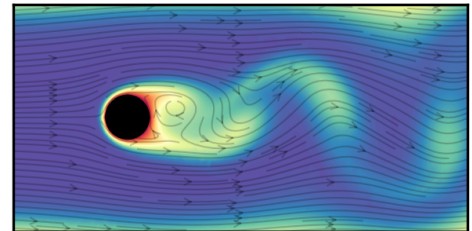

Figure 8: Two snapshots of the passive scalar concentration for the flow past a cylinder.

**Data** Produced in-house with JAX MD. Both the dataset and the script to produce new trajectories will be released. The dataset consists of $10^5$ snapshots: 70% used for training, 10% for validation, 20% for testing.

**Optimization** Adam with learning rate $10^{-3}$. Other parameters are the predefined values in `Optax`'s implementation. Batch size: 8192. 500 epochs. Training time is $\approx 2$ mins.

**Architecture** Multi layer perceptron with CeLu activation function. Dimension of the hidden layer: `[32, 64, 128, 128, 64, 4]`.

**Hyperparameters** Metric loss coefficient $\gamma = 50$.

### F.4   ORDERED MNIST

**Data** out of the full MNIST dataset we generated a trajectory of 1000 steps as discussed in the main text, and evaluated the forecasting accuracy over 1000 different test initial conditions. In Tab. 4 we report the training time for DPNets, DPNets-relaxed and the baselines used. We observe that both our methods are the fastest during training. See also Fig. 10 for the generated sequences of digits by the compared methods.

**Optimization** Adam with learning rate tuned as explained in the general remarks. Batches of 128 samples trained over 150 epochs.

**Architecture** `Conv2d[16]` → `ReLU` → `MaxPool[2]` → `Conv2d[32]` → `ReLU` → `MaxPool[2]` → `Linear[5]`.

For the Auto-Encoder baselines we used this architecture for the encoder and the "reversed" network constructed with transposed convolutions for the decoder.

**Hyperparameters** Metric loss coefficient $\gamma = 1$.

Table 4: Ordered MNIST Training times. Each model is run on CPU and we report mean ± std for 20 runs.

| Model | Fit Time (s) | Time per Epoch (s) |
|---:|:---:|:---:|
| DynamicalAE | $0.571 \pm 0.034$ | $0.057 \pm 0.003$ |
| Oracle-Features | 0.098 | - |
| DMD | 0.333 | - |
| KernelDMD-Poly3 | 4.914 | - |
| KernelDMD-AbsExp | 0.832 | - |
| KernelDMD-RBF | 0.776 | - |
| **DPNets** | $0.046 \pm 0.004$ | $0.049 \pm 0.001$ |
| **DPNets-relaxed** | $0.043 \pm 0.004$ | $0.051 \pm 0.004$ |

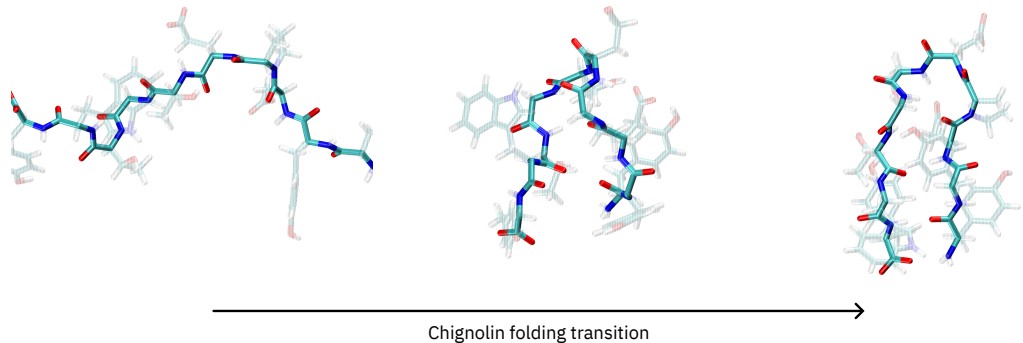

Chignolin folding transition

Figure 9: Three snapshots of Chignolin while undergoing a folding transition.

### F.5 THE METASTABLE STATES OF CHIGNOLIN

In this experiment we build from the work in (Ghorbani et al., 2022) by employing our framework to learn the leading eigenfunctions associated to the dynamics of Chignolin, a folding protein, from a $106\,\mu$s long molecular dynamics simulation sampled every $200\,$ps, totalling over half a million data points (Lindorff-Larsen et al., 2011). We consider every heavy atom for a total of 93 nodes as well as a cutoff radius of 6 Angstroms giving an average of 30 neighbours for each atom. Compared to (Ghorbani et al., 2022), which selects only the $\approx 20\ C_\alpha$ atoms each with its first 5 neighbours, our experiment has therefore a much larger scale. Indeed, (Ghorbani et al., 2022) reports being able to train directly with the objective $\mathcal{P}^0$, while in our case we always encountered numerical errors, and we were able to only succesfully train the $\mathcal{S}^\gamma$ objective. We parametrized the feature map with a graph neural network (GNN) model. GNNs currently are the state of the art in modeling atomistic systems (Chanussot et al., 2021), and allow one to elegantly incorporate the roto-translational and permutational symmetries prescribed by physics. Specifically, we train a SchNet (Schütt et al., 2019; 2023) model with 3 interaction blocks, where in each block the latent atomic environment is 64-dimensional and the inter-atomic distances used for the message-passing step are expanded over 20 radial basis functions. After the last interaction block, each latent atomic environment is forwarded to a linear layer and then aggregated via averaging. The model has been trained for 100 epochs with an Adam optimizer and a learning rate of $10^{-3}$. We analyzed the eigenfunctions using the technique described in (Novelli et al., 2022), which links each metastable state to physically interpretable conformational descriptors. Our analysis aligns perfectly with (Novelli et al., 2022), where the slowest metastable state corresponds to the folding-unfolding transition and is linked to the distance between residues (1, 10) and (2, 9) located at opposite ends of the protein. Additionally, the immediately faster metastable state represents a conformational change within the folded state, characterized by the relative angle between residues 6 and 8.

In Fig. 9 we plot how the structure of Chignolin changes while performing a folding transition, while in Fig. 7 we plot the implied timescales of the dynamical modes of Chignolin as estimated by DPNets-relaxed and Nyström-PCR.

This is by far the heaviest experiment of the paper, and we made use of the package `SchNetPack` (Schütt et al., 2019; 2023) on multiple instances. In particular, we have used `SchNetPack` dataloaders and preprocessing transformations (casting to 32 bit precision and on-the-fly computation of the distance matrix). Further, we have used `SchNetPack`'s implementation of the SchNet interaction block. To reproduce our results, the following informations may prove useful.

**Data** The data was presented for the first time in (Lindorff-Larsen et al., 2011) and is freely available for non-commercial use upon request to DeShaw research. Dataset of 524743 snapshots. Each graph is composed by the 93 heavy atoms. The edges are formed only if two atoms are less than 5 Angstroms distant. The average number of edges is 28.

**Optimization** Adam with learning rate $10^{-3}$. Other parameters are the predefined values in `Torch`'s implementation. Batch size: 192. 100 epochs. Training time: $\approx 11$ hrs.

**Architecture** SchNet with 3 blocks, 20 RBF functions expansions, 64 latent dimension. At the output of SchNet, the hidden variables associated to the nodes are averaged and forwarded to a dense layer with 16 final output features.

**Hyperparameters** Metric loss coefficient $\gamma = 0.01$. For the Nyström baseline we used $M = 5000$ inducing points.

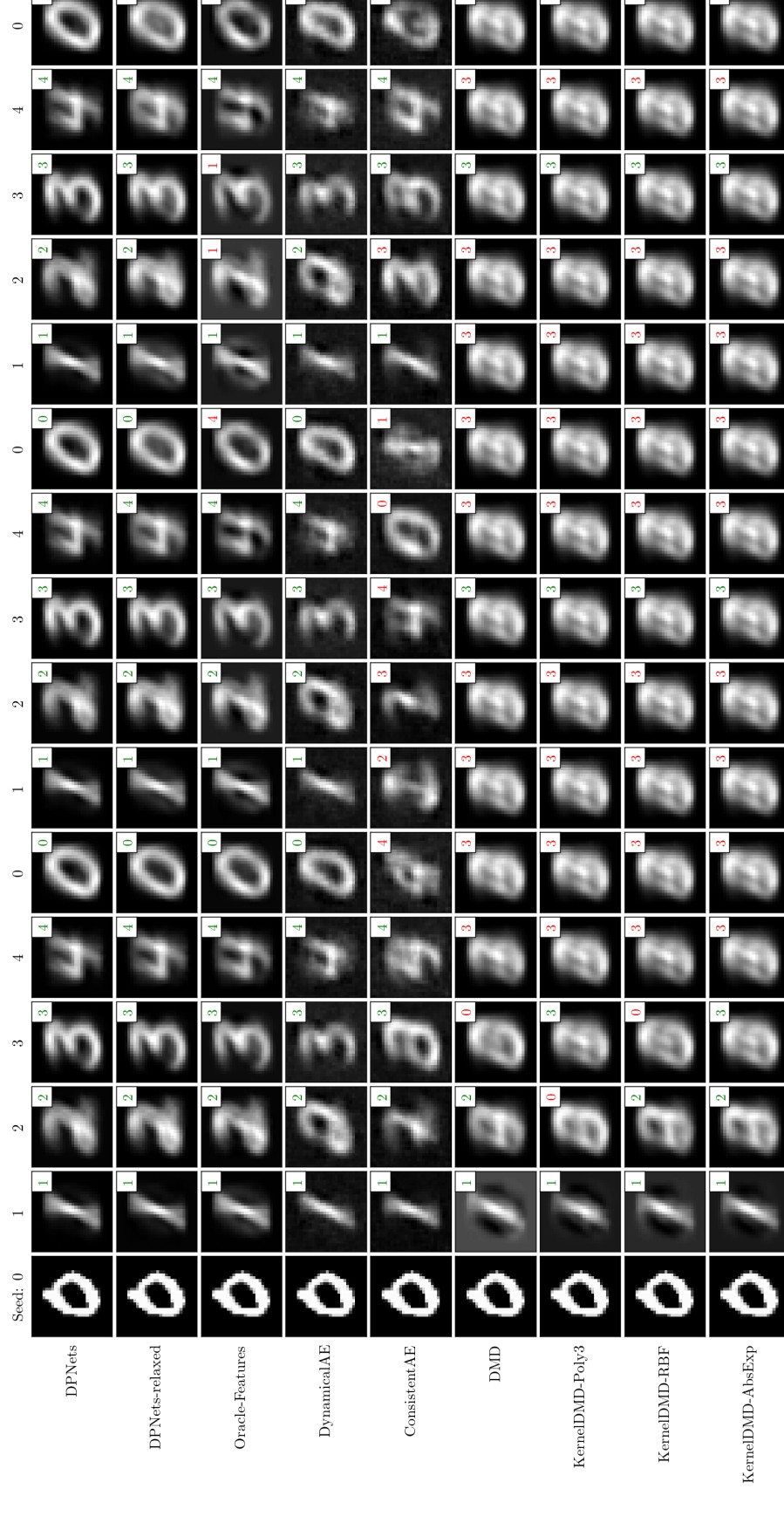

Figure 10: Sample of the states predicted by the different models starting from a seed image of the digit 0. The green label in the upper right corner of each image indicates proper classification, while the red one means miss-classification.