# OpenReview forum: "Learning invariant representations of time-homogeneous stochastic dynamical systems"
_ICLR.cc/2024/Conference — ICLR 2024 poster_

### Official Review · Reviewer_UfpG · 2023-10-31

**Soundness:** 4 excellent
**Presentation:** 2 fair
**Contribution:** 3 good
**Rating:** 6
**Confidence:** 3

**Summary:**

The authors provide a method to learn an invariant representation for dynamical systems helps give an approximation of the transfer operator. They do this while also alleviating weaknesses of previous approaches, such as metric distortion. To do this, the authors show that the search for a good representation can be cast as an optimization problem over the space of neural networks. They provide and an objective function, and tractable approximations to it. They compare their method to other state-of-the-art approaches.

**Strengths:**

- The approach seems novel and interesting enough.
- The authors list the weaknesses of previous approaches (e.g. the weaknesses of previous methods on page 3, under "Learning transfer operators") and show how their method attempts to alleviate them.
- While the authors don't optimize over the main objective, equation (4), they provide tractable approximations, and also qualify and quantify when these approximations are good (e.g. Theorem 1, Lemma 3 in Section C of the supplementary materials).

**Weaknesses:**

- The organization and the explanation of the method could be more straightforward. For example, Algorithm 1 is rather terse, and a more expanded version would be instructive to the reader. It's hard to get birds-eye view of the method because the explanation is spread throughout the text. Perhaps a separate section in the supplementary materials that gives a more direct explanation of the method would help the reader.
- The method requires computing covariance matrices of neural network outputs, which could affect the computation if the output dimension is high.
- I have some reservations about the experiments:
    - In the beginning paragraph of section "5 Experiments", the authors state they only optimize over the learning rate to keep things fair, but I can't seem to find the set of learning rates that were optimized over.
    - On the previous point, the authors state at the top of page 7 that the mini-batch size should be sufficiently large to mitigate the impact of biased stochastic estimators. But, some neural network models benefit from the mini-batch stochasticity of SGD, as opposed to full-batch normal gradient descent. In this case, I would have liked to have seen optimizing over the batch sizes as well, and not just the learning rate.
    - I'm also not sure if optimizing over just the learning rate this is completely fair. I would have liked to see optimizing over more hyperparameters, or at least in addition to the current experiments, to also provide the performance evaluation of the original model implemented in the paper, if possible. For example, in the fluid flow example, Azencot et al. (2022) has a similar fluid flow example, but uses a different architecture and I would have liked to see this one in comparison to the author's method.
    - In the molecular dynamics example, I see Ghorbani et al. (2022) have done similar experiments, but at a smaller scale. Even though the authors performed their experiment at a larger scale, I would have liked to have seen a comparison with Ghorbani et al. (2022) at the smaller scale, or even at the larger scale. This would allow comparing accuracy and training times.

**Questions:**

**Questions:**
- It seems in most experiments, the output dimension of the neural networks, $r$, is low. Would this be the case for most practical applications?
- On page 7 in "Downstream tasks", in order to make predictions of the next step, the authors need to compute: $(\hat{\Psi}_{w}^\top)^\dagger$, i.e. the pseudo-inverse, is that right? If so, it seems this would slow down the computation of the predicted next steps, so how do the computation times compare with other methods for predicting next steps? Any numerical instabilities?
- In Algorithm 1, the authors use $\hat{\mathcal{S}}^\gamma_m$, implying this is the relaxed version, is that right? If so, this isn't stated anywhere in Algorithm 1. Perhaps the authors should have two Algorithms that gives the regular DPNets and also DPNets-relaxed.
- When training the unrelaxed DPNets, how was $\mathcal{P}^\gamma$ evaluated? It is stated in page 5 that in general the covariances are non-invertible and thus $\mathcal{P}^\gamma$ is non-differentiable during the optimization. Also, as the authors state, the use of the pseudo-inverse can introduce numerical instabilities. How much of a problem is this, in your experience? I see that some experiments were unable to use unrelaxed DPNets because of this.
- For Figure 3, in the upper panel - "Ordered MNIST accuracy", I'd imagine that the "Reference" (dashed-line) is just random guessing, i.e. $1/5 = 0.2$? I don't see this stated anywhere, and would like it explicitly stated.
- For Figure 3 in the lower panel, "Fluid Flow RMSE", how many prediction steps were taken? Namely, what are the values of the $x$-axis?

**Remarks:**
- On page 5, on the line above equation (8), the authors say "we show in Lem. 3 in App. B", but I think you mean Section C in the appendix/supplementary materials.
- On page 6, in "Learning the representation", might also need to state that we are also given $\psi'$, i.e. " In practice,given a NN architecture $\psi$ and $\psi$'. "

---

> ### Author Response · Authors · 2023-11-15
>
> We gratefully thank the reviewer for their review and comments, which we now address.
>
> Weaknesses:
> - As you suggested, we have included an expanded version of Algorithm 1 in the supplementary section. Furthermore, upon completion of the review process, we intend to publicly share the GitHub repository containing not only the method's implementation but also the implementation of each experiment.
> - Thank you for raising this important point. Please see our global reply where we address this point in detail, as well as our answer to reviewer **hdvQ** for further comments on the computational complexity
> - Experiments:
> 	- The grid used for the search of the best learning rate was reported in appendix F: 100 equally spaced values in the interval $(10^{-6}, 10^{-2})$. To aid reproducibility, in the updated version we also added a table with the learning rate found by the search for each (experiment, method) pair.
> 	- We believe this is a valuable suggestion, and while we keep the batch-size for DPNets and VAMPNets as high as possible to not incur any bias upon estimation of the score function (see our discussion in Sec 4), we are in the process of re-running the Auto-Encoder methods with smaller batch-sizes. We will give an update on this point as soon as the new results are available.
> 	- Given the number and diversity of the baselines (each with a different set of hyperparameters) the evaluation methodology we devised is to fix every parameter defining the model expressivity (architecture, feature-dimension) and train the best possible model within a budget of epochs. Hence we optimized learning rates and (thanks to your suggestion) batch sizes. Taking a closer look at the fluid flow example in Azencot et al. (2022), one can see that our setting is remarkably close to theirs: almost identical underlying PDE, same Reynolds number, as well as similar size of the dataset. With the exception of a different activation function, we also use a similar MLP architecture (though the one in our experiment is slightly bigger, with 3.8 million parameters compared to theirs which is 2.49 millions). Finally, our implementation agrees with their findings, for which in this experiment they expect results in line with dynamical AEs (see our Fig. 1).
> 	- The experiment in Ghorbani et al. (2022) considers different proteins. Their method (which is equivalent to VAMPNets) fails on the larger scale Chignolin experiment as we report in Sect. 5. On the other hand applying our method to their smaller scale dataset should be doable, although the hyperparameter tuning may be time consuming. While we cannot guarantee to have it done in the time span before the end of the rebuttal period, we will definitely include this experiment in the revised version.
> Questions:
>
> - The answer is yes. Please see our global reply for an in-depth discussion of the computational complexity of using covariances.
>
> - That formula is just for definition purposes. In practice we simply compute it as $(\widehat{\Psi}^\top_w)^\dagger \widehat{F}’ =( \widehat{\Psi}^\top_w \widehat{\Psi}_w)^\dagger \widehat{\Psi}_w^\top  \widehat{F}’$, which is equivalent to solving $\ell$ least squares with $r\times r$ matrix $\widehat{\Psi}^\top_w \widehat{\Psi}_w$. Finally, notice that this is performed only once when fitting the operator regression model. Once computed it can be stored and used to predict future states through simple matrix multiplications. Additionally, low rank estimators, as well as Nyström based methods are available to considerably speed up this initial fitting cost even when $r$ or $n$ are very large (see Meanti et al. 2023).
> - In Section 4, just before the “Operation regression” paragraph we actually state: In Alg. 1 we report the training procedure for DPNets, which can be also applied to the other scores (8) and (14) in step 4, respectively. When using mini-batch SGD methods to compute ∇ ̂ Pγ m(w) or ∇ ̂ Sγ m(w) via backpropagation at each step, m ≤ n should be taken sufficiently large to mitigate the impact of biased stochastic estimators”. As you point out, the algorithm is for DPNets-relaxed, while in the main text we stated that it is for DPNets. We have now corrected this.
> - $\mathcal{P}^\gamma$ was evaluated using `torch.pinv` with the “hermitian” flag set to True. The problem of using pseudo-inverses lies in the backpropagation step, as the pseudo-inverse is known not to be differentiable whenever the rank is not stable over the parameter space (see Golub & Pereyra, 1973). On the other hand, on the forward pass, the torch implementation seemed to be stable enough, and when we weren’t able to train DPNets (unrelaxed) and VAMPNets it was a failure of the back-propagation step.
> - This is right, we are now stating it explicitly, thank you.
> - 40 prediction steps, that is the full test set.
>
> Remarks:
> - That’s right. We have now fixed it, thank you.
> - Absolutely. Fixed it as well in the newly uploaded draft.

---

> > ### Author Response · Authors · 2023-11-19
> > **Batch-size optimization**
> >
> > Dear Reviewer, we finished the re-evaluation of the experiments with batch-size optimization (jointly with the learning rate optimization). Both Consistent and Dynamical AEs didn’t show any changes. Specifically, the results of 20 independent runs at different batch sizes are still within the confidence intervals of the results already reported in the original manuscript. Clearly, upon optimizing the batch sizes, we have re-optimized the learning rate too.
> > As a follow up to your question we have also done a little literature search on the topic of optimizing the batch size. A relevant paper on the topic is Shallue et al. 2019  “Measuring the Effects of Data Parallelism on Neural Network Training”, studying the interplay between the batch size and the number of training steps required to reach a given target metric. Even if increasing the batch size eventually leads to diminishing returns in terms of number of training steps, according to this work there is no convincing evidence that tuning the batch size has any effect on the test performance of a model. We hope this fully answers your question. We definitely learned something new!

---

> ### Comment · Reviewer_UfpG · 2023-11-21
>
> I thank the authors for their detailed response, revisions, and additional experiments. I have read the other reviews and the discussions that followed (as of this writing).
>
> My questions and concerns have been mostly addressed. Thank you for the reference, although I'll note that in reviewing the reference and in the context of the submission, they mention when training in large batch regimes, it may help to tune other hyperparameters as well, as they can differ from their small batch regimes (e.g. learning rate, momentum, etc.). In fact, in Section 4.7 of the reference [(Shallue, 2019)](https://jmlr.org/papers/volume20/18-789/18-789.pdf) they explicitly state:
>
> > "We also found that the best effective learning rate should be chosen by jointly tuning the learning rate and momentum, **rather than tuning only the learning rate.**" [emphasis mine]
>
> So it may help to also tune the momentum if the authors also believe that will be productive. But I do recognize and sympathize that hyperparameter tuning can be computationally expensive.
>
> But my current thoughts are: after reading the response and the other reviews, and due to the limited gradations of the scoring system (score would go up if there were sufficient gradations), I am inclined to have my score remain unchanged. I acknowledge the hard work put into the rebuttal and revisions, and am very grateful and extremely appreciative to the authors.

---

> > ### Author Response · Authors · 2023-11-22
> > **Thanks for the feedback**
> >
> > Thank you for engaging in the discussion, your feedback and appreciation of our rebuttal.
> > We think that both your remarks in the review and the following discussion helped us communicate better our contributions.

---

### Official Review · Reviewer_1QLL · 2023-10-31

**Soundness:** 3 good
**Presentation:** 3 good
**Contribution:** 2 fair
**Rating:** 6
**Confidence:** 4

**Summary:**

In this paper a particular representation of a state space of a dynamical system is learned.
The representation is intended to correspond to the subspace of highest singular values of the transfer operator of the system, of a fixed predifined dimension. It is learned from the observed trajectories of the system and parametrised by neural networks.

The corresponding optimisation objective can be expressed in terms of the covariance matrices of the data (in the representation space).  While the direct involves matrix inversion and is computationally difficult, a proxy objective, which is proven to be a lower bound, is proposed. The proxy objective has better computational properties.

Once the representation is learned, the transfer operator can be learned using standard approaches based on operator regression.  Experiments are performed to demonstarte a competitive performance of the approach.

**Strengths:**

This paper is clearly written and the topic of modeling dynamical systems is important.
The proposed representation space is very natural.

**Weaknesses:**

This is mainly an empirical paper, as it proposes to model a certain space using neural networks with gradient descent optimsed objective.  However, the empirical evaluation of the method is not completely convincing, and some points are not clear.


In addition, even the relaxed objective introduced in the paper is computationally heavy. It involves
computing the covariannce matrices as _differentiable functions_. This means the computation graph involves summing $n$ matrices of size $r \times r$, each coordinate of which is a differnetiable function.
As noted in the paper, minibatches could be used but would introduce bias. Even with minibatches, the
representation dimension $r$ would have to stay low, and appears to be low in all experiments (in which it was specified).


A more detailed notes on experiments:

**Ordered MNIST:** My understanding is that this example is new in this paper. Is there an explanation/intuition why other methods fails here? Are convolutional networks used to model the
respresentation? I assume the non NN methods fail simply since representing images is easier with convolutions. Why do other methods fail? I believe explaining the difference here could make the paper considerably stronger.


**Logistic Map:** The takeaway here is not clear.  DPNets has the worst spectral error among all methods.
In addition, there appears to be no theoretical reason why DPNets-realxed should perform better on this metric, as it was introduced as a computational relaxation. Do the suggest it has properties allowing it to be a better estimator in general?

**Continuous dynamics:**
The scale of Fig. 2 lower pane is that of 200. This makes is very hard to understand whether the smaller
eigenvalues are estimated accurately and what is the order of magnitude.  Displaying ratios estimated/true would be more informative. Also, are there other ways to estimate the eigenvalues?


**FlowRMSE:**
The general magnitude of the quantities in this experiment is not clear.  If the quantities are of order 10, then the difference between errors of order 0.1 and 0.001 is less impactful, and all methods perform well.
What are the typical values of the quantities for which RMSE is computed? Again, percentage ratios would be much clearer than absolute values.



**Chignolin Experiment:**
 In Table 2, describing the Chignolin experiment, while estimated quantities are somewhat closer to the reference values than the true quantities, they still look quite far. It is thus not clear what is the point of this part of the experiment.
The second part of that experiment, involving Fig.4, unfortunately can not be understood from the main text. (what is the free energy surface, what is known about its proper minima, why matching the results of Bonati et al   is of interest? ).

**Questions:**

In addition to the points above:

Is the relaxation objective (9) a new contribution of this paper, or were similar relaxations employed in previous work (for instance VAMPSNets, DeepCCA)?

---

> ### Author Response · Authors · 2023-11-15
>
> We gratefully thank the reviewer for their review and comments, which we now address.
>
> While we address the specific comments on the empirical evaluation of the methods below, we respectfully disagree on the point that the paper is “mainly empirical”. Our aim is to provide a method firmly grounded in state-of-the-art statistical learning theory, as evidenced by our detailed discussions in Sec. 2 and 3, as well as App. A–E, outlining the theoretical foundations of DPNets. In reply to your question, the metric distortion loss $\mathcal{R}$, as well as the relaxed score in eq. (9) haven’t been proposed before and are novel contributions. We will be happy to discuss any theoretical question/comment, should it arise during the rebuttal period.
>
> The point on the high cost of evaluating covariances is important, and we are grateful to the reviewer for raising it. In this respect, we refer to the general answer for a detailed account on why we expect that in many scenarios a low dimension to be sufficient and what one can do if, for any reason, the representation dimension needs to be large.
>
> We also refer to point 1 of our answer to reviewer **hdvQ** for further comments on the computational complexity.
>
> On the specific experiments:
>
> **Ordered MNIST**: this example was actually introduced in Kostic et al. 2022. In the revised version of the manuscript we now acknowledge it, thanks for pointing it out. In our DPNets methods as well as in VAMPNets, oracle-features, and in the AE baselines we use the same CNN (the specific architecture is reported in App. F.4), and as you said non-NN methods fail because representing images is easier with convolutions. VAMPNet, which is the closest to DPNet in terms of methodology, fails to converge (see our discussion on its instability in Sec 3). The DynamicalAE actually learns a reasonable representation, making it the fourth best model behind DPNets and the oracle-features (which, however, are trained using also the true labels). Upon visual inspection, indeed, DynamicAE returns readable images (see the Figure on the very last page of the Appendix). Finally, we speculate that ConsistentAE does bad in this setting as they also enforce a “consistency” constraint for the time-reversed system, which is not satisfied by the true dynamics of this example. Finally, even though oracle-features have been trained using the true labels, they are oblivious of the dynamics of this system. This is why, even attaining very good performances, they eventually fail at long term prediction.
>
> **Logistic Map**: We do not have any indication that DPNets-relaxed provides a better estimation in general. Indeed, for larger feature dimensions, the unrelaxed DPNets is on par with DPNets-relaxed. To clarify the takeaway of this experiment, we significantly expanded the App. F.1 with an in-depth discussion of the inherent difficulty when estimating the spectra (high sensitivity due to nonnormality of the operator) and the role of the dimension of the feature map, which is particularly important in this context.
>
> **Continuous dynamics:** For improved readability we have changed the scale to logarithmic, and plotted the implied transition rate, that is the absolute value of the non-null eigenvalues. We are only aware of the alternative methods presented in (Klus et al. 2019) and (Hou et al 2023), which correspond to DMD and kernel-DMD, respectively, adapted for learning the generator. Our example corresponds to using the DMD method coupled with a DPNet representation.
>
> **Fluid flow**: We completely agree with your comment, and in the updated draft we plot the results in terms of ratios as you suggest. Working with RMSE, we find it sensible to use the standard deviation of the data (which is approximately ~ 0.1675) as a proxy for the general magnitude of the quantities involved.
>
> **Chignolin**: Indeed, quantities such as the free energy are very specific to the context of physical sciences. In the revised version of the draft we added an explanation of what free energy is, and how it relates to the metastable states. Concerning your question about the estimated quantities being far from the true ones, it basically boils down to the amount of data at our disposal. Molecular dynamics simulations are great in that they allow to simultaneously probe the dynamics of a physical system, as well as their equilibrium properties if the simulation is long enough. For big systems, however, to accurately estimate equilibrium quantities such as the transition time or the enthalpy one needs extremely long simulations, often out of reach. The trajectory that we used, while large scale, is simply not long enough to entirely capture the equilibrium properties with high accuracy. We point out that speeding up molecular dynamics simulations is an area of active and intense research, and algorithms to estimate the equilibrium properties from short, imperfect simulations are highly valuable.

---

### Official Review · Reviewer_sgxX · 2023-11-01

**Soundness:** 3 good
**Presentation:** 2 fair
**Contribution:** 2 fair
**Rating:** 6
**Confidence:** 3

**Summary:**

This paper presents a relatively new approach to learning the operator for dynamical systems. Instead of one-step modeling of the neural operator, the authors suggest first learning the invariant representations, and then leveraging these learned representations to perform operator regression. The authors assessed their results on various datasets, including an ordered MNIST, a 2D fluid flow, and a molecular dynamics simulation, primarily using metrics such as spectral error and RMSE.

**Strengths:**

This paper provides a new way of learning invariant representations of dynamical system.
The authors provide theoretical justifications for the derivation of their objectives functions.

**Weaknesses:**

W1. The pipeline design suggests that there might be a possibility of error accumulation from the first phase of learning representations to the second phase of learning dynamics. It would be beneficial if the authors could provide empirical justification to show that their design is comparable to or better than learning the neural operator in a single step. For instance, establishing baselines using Frameworks like FNO or Deeponet would be ideal.

W2. Tying in with W1, the efficacy of the proposed method, particularly the operator regression in the second part, remains uncertain when applied to chaotic systems where acquiring accurate representations in the initial phase may prove challenging

W3. It would be great if the authors could evaluate the performance on insightful metrics like Lyapunov spectrum. And including discussion on recent related works (e.g., https://arxiv.org/pdf/2304.12865.pdf) should be necessary.

W4. If the primary goal is representation learning, I feel that discussions and comparisons related to popular representation learning methods (e.g., contrastive learning) should be discussed and included.

**Questions:**

Q1. I am curious about the empirical performance of the results concerning different hyperparameters, particularly the value of $r$.

Q2. Additionally, what would be the loss for the OLS estimator of the transfer operator across different datasets? It would be concerning if the performance is highly sensitive to the value of $r$.

Q3. For the dynamical data, i.e., the 2D fluid flow and the molecular dynamics, it would be beneficial if the authors could provide the visualizations of the dynamics.

---

> ### Author Response · Authors · 2023-11-15
>
> We gratefully thank the reviewer for their review and comments, which we now address.
>
> W1. We recall that, among the baselines we analyze, both KernelDMD, DMD, DynamicalAE and ConsistentAE learn the transfer operator in a single step. Indeed, the first two do not require to pre-learn a representation, while the latter two optimize a sum of losses encompassing both representation (i.e. the reconstruction error of the AE) and dynamics. As explained in Sect. 2, and specifically in the paragraph “Learning transfer operators”, off-the-shelf representations often used alongside DMD-alike methods are detrimental in that they induce metric distortion as well as representation biases. In contrast, DPNets are designed to mitigate these effects. Further, the discussion around Eq. (3) entails that for all compact transfer operators, the cumulative error of DPNets plus consistent operator regression is controlled; This is also what we empirically show in every experiment. Does the reviewer agree on this point? We’d be happy to clarify any aspect of our response if needed.
>
> Regarding establishing baselines using FNO/Deep-O-net, we would like to highlight that these techniques are solving a different problem. Although the concept of “neural-operator” is quite broad, the methods you mentioned have been specifically designed to learn surrogate maps for the solution operators of PDEs; see (Kovachki et al., 2023). Moreover, these methods often require knowledge of the underlying physical equations to generate the training data, which is something not at hand for general stochastic processes. Apart from the fluid flow example, for which we already included a physics-informed baseline, the applicability of FNO/Deep-O-net to solve the problem of transfer operator learning remains unclear. We are open to further discussion on this aspect if the reviewer deems it necessary.
>
> W2 and W3. Ergodic theory is usually studied using the geometric approach or the operator theoretic approach. While the first is related to the study of Lyapunov exponents and fractal dimension, the latter is related to the Transfer/Koopman operators of Markovian processes. Connecting these two complementary approaches to ergodicity is an ongoing research, see e.g. (Tantet et al. 2018, Brunton et al  2017; 2022, Mezić and Avila 2023). Due to the nature of our work, and the central role played by stochastic dynamical systems, we have focused on the spectrum of the transfer operator as relevant information about the dynamics. On the other hand, the Lyapunov spectrum is much more interesting in the context of deterministic, and especially chaotic systems.  We can elaborate on both of these approaches in Sec. 1 also mentioning (Platt et al. 2023) as an example of deep learning methods for deterministic systems based on the Lyapunov spectrum approach. Would the reviewer consider this discussion appropriate and to be included in the revision? Finally, concerning the synthetic experiment of Noisy Logistic map we can report that, in the noiseless case the Lyapunov exponent is log(2), while introducing the noise in the model yields the Lyapunov exponent (in the stochastic sense) equal to 0, see (Baxendale, 1989).
>
> W4. Since representation learning is a broad field, in the “Previous work” paragraph we only covered representation learning in the context of transfer operator (Azencot et al., 2020; Lusch et al., 2018; Morton et al., 2018; Otto and Rowley, 2019). We are not aware of contrastive learning approaches in the context of dynamical systems. However, we welcome any suggestion on additional references to be included in the discussion.
>
> Q1. In the revised version of the manuscript (Appendix F.1) we added an in-depth discussion on the role of the feature dimension in the Logistic map experiment. It is also interesting to note that DPNet showed to be a quite “parsimonious” representation, needing only a small dimension to well approximate the dynamics. This is particularly relevant also on the ordered MNIST example, where a feature map of just 5 floats achieves > 98% accuracy on validation.
>
> Q2.  Our DPNets method learns a representation of dimension $r$ and subsequently, an OLS operator estimator can be trained using this representation. To evaluate the quality of the learned $r$-dimensional representation it is not meaningful to look at the OLS error (or risk) but rather to the forecasting error or the spectral error, which we have reported in Q1 for the logistic map. Indeed, although the OLS error seems the natural object to look at, it does not reflect the quality of the representation. For instance, a trivial representation mapping every state to zero would have a zero error. We’d be happy to further clarify any aspect of our response if needed.
>
> Q3. We have added visualizations for both experiments (see Figures 8 and 9).

---

> > ### Author Response · Authors · 2023-11-15
> > **Additional references for our original answer**
> >
> > Kovachki, N. B., Li, Z., Liu, B., Azizzadenesheli, K., Bhattacharya, K., Stuart, A. M., & Anandkumar, A. (2023). J. of Machine Learning Research, 24(89), 1–97.
> >
> > Brunton, S.L., Brunton, B.W., Proctor, J.L. et al. (2017 ) Chaos as an intermittently forced linear system. Nature Commun 8/19.
> >
> > Mezić, I. and Avila, A. M., (2023) Spectral Properties of Pullback Operators on Vector Bundles of a Dynamical System, SIAM Journal on Applied Dynamical Systems 22/4
> >
> > Tantet, A., et al. (2018) Crisis of the chaotic attractor of a climate model: a transfer operator approach. Nonlinearity 31.55
> >
> > Baxendale P. H. (1989) Lyapunov exponents and relative entropy for a stochastic flow of diffeomorphisms,  Probability Theory and Related Fields

---

> ### Comment · Reviewer_sgxX · 2023-11-23
> **Thank you for your detailed reply**
>
> I want to thank the authors first for their detailed reply. I acknowledge the authors' replies regarding my W1 & W2. And I appreciate the authors provided additional experiments and visualizations for my Q1 & Q3. I still hold hesitations towards related works and notice Reviewer 1QLL's question regarding the comparison of the objective function against VAMPNets and DeepCCA (for which I have not found the authors' response).
>
> An update: After careful rethinking and as we are approaching to the closing of the discussion period, I decided to change my score to 6. I feel in general impressed by the authors’ hard work. And I hope further clarifications regarding the novelty in terms of the objective function comparing against previous works, as well as the concerns regarding the computational cost would be added into the revision.

---

> > ### Author Response · Authors · 2023-11-23
> > **Thanks for your feedbacks**
> >
> > Thank you for engaging in the discussion, your feedback and appreciation of our rebuttal.
> > We will include all your remarks, which we believe provide a broader context for our method within the relevant literature, in the next revised version of the manuscript. In particular, we will expand the related work paragraph, highlighting alternative approaches to transfer operators (e.g. Neural operators, PINNs, and Deep-O nets), and we'll compare learning stochastic vs deterministic dynamical systems, specifically mentioning chaotic systems.
> >
> > We will also clarify the novelty of our objective function in the Contributions paragraph within the introductory section.
> >
> > Finally, regarding the discussion on the computational cost in App. D.4, now improved in the revised manuscript, we agree that it is very relevant and we plan to highlight it also in the main body of the paper.

---

### Official Review · Reviewer_hdvQ · 2023-11-09

**Soundness:** 3 good
**Presentation:** 3 good
**Contribution:** 3 good
**Rating:** 6
**Confidence:** 2

**Summary:**

This paper proposes a method for learning a representation of continuous and discrete stochastic dynamical systems that achieves state of the art results on many datasets.

**Strengths:**

1. The paper identifies the learning problem as an optimization problem and provides an efficient way to solve it. To overcome singularity, they have a relaxed score function and they also prove that the relaxed score has theoretical guarantees. The proof is mathematically solid.

2. Many experiments are done to verify their claims with impressive results.

**Weaknesses:**

1. The computation for covariances for $\psi_{w_j}$ and $\psi'_{w_j}$ might be costly. How does this computation cost compare to the previous methods? Is there a way to speed up?

2. For the continuous-time dynamic, the score function might be unstable. Have you observed this in experiments?

**Questions:**

See weaknesses

---

> ### Author Response · Authors · 2023-11-15
>
> We gratefully thank the reviewer for their review and comments, which we now address.
>
> 1. This point was raised by different reviewers and we refer to the general response for a detailed discussion. Concerning the more specific questions about the computational complexity:
> 	1. **DPNets**: the computational complexity for both the forward and backward pass is exactly the same of **VAMPNets**, as the addition of the metric distortion loss $\mathcal{R}$ (see also Eq. 8) is actually of $\mathcal{O}(r^2)$, $r$ being the dimension of the representation. Thus, when $ r > n$, the computational complexity is dominated by the (pseudo-)inversion which scales as $\mathcal{O}(r^3)$. Kernel-based operator regression algorithms such as DMD also share the same (forward) computational complexity, requiring the evaluation and inversion of a covariance ($\mathcal{O}(nr^2 + r^3)$) or kernel matrix ($\mathcal{O}(n^3)$). These algorithms do not-require backpropagation, but often fall short in representation power — as shown in the experiments.
> 	2. **DPNets-relaxed**: The computational complexity of the relaxed score is actually _lower_ than that of DPNets and VAMPNets, since it does not require the full inversion of a matrix, but only its leading singular value (i.e. its operator norm), which can be computed via standard Arnoldi/Lanczos methods, yielding a total complexity of just $\mathcal{O}(nr^2)$.
> 2. The optimization of the score in the Langevin dynamics experiment did not show any signs of instability. We speculate that this is due to both the low-dimensionality of the state space and the infinitely-smooth ($C^{\infty}$) potential function. Indeed, since the score for continuous dynamical systems is the sum of the (finite) eigenvalues of the symmetric eigenvalue problem $C^w_{X\partial} - \lambda C^w_X$, in non-pathological cases, its value and it derivatives can be computed efficiently in a numerically stable way, see e.g. (Andrew, A. L. and Tan, R.C.E., Computation of Derivatives of Repeated Eigenvalues and the Corresponding Eigenvectors of Symmetric Matrix Pencils, SIMAX 20/1, 1998), ). We have included this remark in the revised manuscript. Notwithstanding these favorable conditions, we already acknowledged the potential instability of the score for continuous dynamics in the conclusions. While beyond the scope of the paper, an intriguing avenue for future research involves an in-depth analysis of continuous dynamics.

---

> > ### Comment · Reviewer_hdvQ · 2023-11-22
> >
> > Thank you so much for addressing my questions. After careful consideration, I decided to keep my rating unchanged.

---

> > > ### Author Response · Authors · 2023-11-22
> > > **Thanks for your feedback**
> > >
> > > We once again thank the reviewer for their feedback. If there are any last-minute questions or comments, we would be glad to address them.

---

### Author Response · Authors · 2023-11-15
**Global response**

Dear AC and reviewers, in this global response we summarize the main revisions made to our paper following the first round of reviews. We updated the paper in OpenReview, highlighting new material in red. Moreover, we address a key point raised by several reviewers about the computational complexity of our method.
We would also like to thank the reviewers for their useful feedback and suggestions.

**Paper improvements highlights:**

1. We improved Figure 2-3: we plotted the estimated transition rates for the Langevin dynamics and the relative RMSE error for the fluid dynamics for better readability.
2. We expanded the supplementary material adding:
    1. visualizations of the molecular dynamics and fluid dynamics,
    2. the learning rates for each experiment (Table 3 in App. F),
    3. an in-depth discussion (App. F.1) on the role of the feature dimension for the noisy logistic map experiment
    4. a detailed version of Algorithm 1 (Alg. 2 in App. D.4), together with an improved discussion of the time complexity.

We are also working towards optimizing over the batch size (where appropriate) as well as re-running the molecular dynamics experiment with GNNs on a smaller scale dataset to directly compare to Ghorbani et al. 2022. We are doing our best to have these additional results ready by the end of the rebuttal period.

**On the computational costs of using covariances:**

As several reviewers pointed out, our method requires the computation and back-propagation over covariance matrices. This can incur high computational costs when the feature dimension is larger than a few hundreds.

In all of our examples we kept the feature dimension low, and we never experienced any computational bottlenecks.We had good reasons to use moderate feature dimensions: many dynamical systems, like those we considered, evolve within a low-dimensional manifold, hence a proper representation allows for an extreme dimensionality reduction. This is not merely anecdotal evidence, but it aligns with the fact that the transfer operator (which we aim to learn) is often not only compact, but it has an effective low rank structure. For instance, in the ordered MNIST experiment a feature dimension of 5 perfectly captures the dynamics of 28x28=784 - dimensional images. Likewise, in the large-scale Chignolin example,  a feature map of 16 captures the most important physical modes of a protein with more than 100 atoms.

Notwithstanding, sparked by the observations of the reviewers, we asked ourselves what strategies are at disposal when the feature dimension has to be, for any reason, large. If $X$ is the $n \times r$ batch of covariates ($n$ samples $\times$ $r$ features), their covariance is given by $X^T X = \sum_{i = 1}^{n} x_i x_i^T$, yielding the cost $\mathcal{O}(n r^2)$.This computational cost is not uncommon in the deep learning literature. For example, the CCA family of methods in self-supervised learning, e.g. VICReg/BarlowTwins/SWAV/W-MSE, see (Balestriero, A Cookbook of Self-Supervised Learning, arxiv:2304.12210v2), have the same computational complexity. In these methods, computations can be embarrassingly parallelized by splitting the sum over n, as well as its gradient, across multiple GPUs. Finally, given the formal analogy between computing a covariance of $n$ examples of $r$ dimensions to computing the attention matrix of a context of $r$ tokens of dimension $n$, one can borrow ideas from the flourishing literature on sub-quadratic attention, see for example: Linformer by Wang et al. 2020,  RFA by Peng et al. 2021 or Performers by Choromanski et al. 2021.

---

### Meta-Review · Area_Chair_gaqD · 2023-12-08

**Metareview:**

The authors present a two-step method for learning the dynamics of stochastic dynamical systems which involves learning the invariants of the system and then learning an operator that acts on the state space with the invariants divided out. The reviewers agreed that the work was valuable and criticism mainly focused on the computational cost of taking the covariance matrix of the features, but for systems where the feature dimension is in the hundreds or less the method can be useful, and further approximations could possibly extend this further. Although the reviews were borderline, the reviewers all seemed satisfied by the authors' response and I recommend acceptance.

**Justification For Why Not Higher Score:**

This was addressed in the metareview.

**Justification For Why Not Lower Score:**

This was addressed in the metareview.

---

### Decision · Program_Chairs · 2024-01-16

Accept (poster)